# Early-life exercise induces immunometabolic epigenetic modification enhancing anti-inflammatory immunity in middle-aged male mice

Nini Zhang[1,2,8], Xinpei Wang [1,8], Mengya Feng[1,3], Min Li[1], Jing Wang[4], Hongyan Yang[1], Siyu He[1], Ziqi Xia[1], Lei Shang[5], Xun Jiang[2], Mao Sun[6], Yuanming Wu [6], Chaoxue Ren[7], Xing Zhang [1], Jia Li [1,5] ✉ & Feng Gao [1] ✉

Exercise is usually regarded to have short-term beneficial effects on immune health. Here we show that early-life regular exercise exerts long-term beneficial effects on inflammatory immunity. Swimming training for 3 months in male mice starting from 1-month-old curbs cytokine response and mitigates sepsis when exposed to lipopolysaccharide challenge, even after an 11-month interval of detraining. Metabolomics analysis of serum and liver identifies pipecolic acid, a non-encoded amino acid, as a pivotal metabolite responding to early-life regular exercise. Importantly, pipecolic acid reduces inflammatory cytokines in bone marrow-derived macrophages and alleviates sepsis via inhibiting mTOR complex 1 signaling. Moreover, early-life exercise increases histone 3 lysine 4 trimethylation at the promoter of *Crym* in the liver, an enzyme responsible for catalyzing pipecolic acid production. Liver-specific knockdown of *Crym* in adult mice abolishes this early exercise-induced protective effects. Our findings demonstrate that early-life regular exercise enhances anti-inflammatory immunity during middle-aged phase in male mice via epigenetic immunometabolic modulation, in which hepatic pipecolic acid production has a pivotal function.

Regular physical activity has gained widespread recognition as an efficacious endogenous "polypill", offering individuals an array of health advantages, including increased fitness levels, improved cardiovascular health and an enhanced overall quality of life[1–4]. In particular, chronic regular exercise training has been shown to positively modulate immune function and confer effective protection against various infections[5]. A growing body of epidemiological evidence indicates that regular physical activity is associated with reduced mortality and incidence of respiratory infections, including COVID-19 infection[6,7]. Moreover, regular exercise has been demonstrated to mitigate chronic inflammation, a condition that may suppress immune function and exacerbate the susceptibility to infections[8].

[1]Key Laboratory of Aerospace Medicine, Ministry of Education; School of Aerospace Medicine, Fourth Military Medical University, Xi'an, China. [2]Department of Pediatrics, Tangdu Hospital, Fourth Military Medical University, Xi'an, China. [3]Center for Mitochondrial Biology and Medicine, Key Laboratory of Biomedical Information Engineering of Ministry of Education, School of Life Science and Technology, Xi'an Jiaotong University, Xi'an, China. [4]Department of Orthopedics, Xijing Hospital, Fourth Military Medical University, Xi'an, China. [5]Key Laboratory of Hazard Assessment and Control in Special Operational Environment, Ministry of Education; Department of Health Statistics, School of Public Health, Fourth Military Medical University, Xi'an, China. [6]Department of Biochemistry and Molecular Biology, Center for DNA Typing, Fourth Military Medical University, Xi'an, China. [7]School of Sport and Health Science, Xi'an Physical Education University, Xi'an, China. [8]These authors contributed equally: Nini Zhang, Xinpei Wang. ✉e-mail: jiali816@fmmu.edu.cn; fgao@fmmu.edu.cn

Emerging evidence suggests that maintaining an active level of physical activity or healthy aerobic fitness during early childhood may have a lasting impact on health outcomes in adulthood. The Amsterdam Growth and Health Longitudinal Study, a longitudinal investigation, has demonstrated that promoting physical activity in adolescence is associated with a range of favorable health outcomes in adulthood, including increased aerobic fitness, increased bone mineral density, lower body mass index, and reduced body fat mass[9]. Similarly, a retrospective study has revealed that engaging in regular physical activity during youth is linked to decreased rates of developing arterial hypertension and type 2 diabetes mellitus in adulthood[10]. Considering the vital role of immune defense against infections in healthy aging[11], there is surprisingly limited research exploring the long-term effects of regular exercise on immune function, particularly the impact of early-life exercise in enhancing immunity against infections later in life.

Growing evidence indicates that metabolism and immunity are closely interconnected, giving rise to a new area of research known as immunometabolism[12]. Previous studies have demonstrated that moderate- and vigorous-intensity aerobic exercise for over 20 min triggers an obvious metabolomic response, especially in relation to various lipid super-pathways, as the body depletes its glycogen reserves[13,14]. Moreover, exercise promotes the catabolism of branched-chain amino acids, which may potentially augment insulin sensitivity and mitigate the risk of type 2 diabetes, along with their effects on glucose and lipids[15,16]. Recent evidence has highlighted the regulatory role of specific intermediate metabolites in immune function. For instance, lactate, a byproduct of glycolysis, has been shown to induce pro-inflammatory phenotype in T cells and anti-inflammatory phenotype in monocytes[17,18]. Conversely, succinate, a Krebs cycle intermediate, acts as a chemical messenger to trigger inflammatory responses[19]. Although it is established that acute exercise promotes the accumulation of lactate and succinate, the long-term effects of exercise on circulating metabolites and immune function have yet to be fully understood.

Here, we show that swimming training in early life curbs the inflammatory response to lipopolysaccharide (LPS), reduces sepsis severity and improves recovery in male mice during middle-aged phase, even after a long interval of detraining. This enduring immunoregulatory benefit is attributed to epigenetic modulation of an immunometabolic network, in which pipecolic acid is identified as an important factor.

## Results

### Early-life exercise alleviates LPS-induced sepsis and promotes recovery in middle-aged mice

The exercise protocol is depicted in Fig. 1a. The overall health of sedentary (Sed) and exercised (Exe) mice was evaluated at 4 and 15 months of age. Compared with Sed mice at 4 months of age, Exe mice that underwent 3 months of swimming training starting from 1 month of age exhibited decreased white blood cell (WBC) counts, including neutrophils, lymphocytes, and monocytes (Supplementary Fig. 1a), and decreased pro-inflammatory cytokines TNF and IL-1β (Supplementary Fig. 1b, c), but increased anti-inflammatory cytokine IL-1Ra (Supplementary Fig. 1d) in the blood. However, after 11 months of detraining, all these indexes showed no significant differences between Sed and Exe mice at 15 months of age (Supplementary Fig. 1a–d).

To evaluate the immune response in acute infection, both Sed and Exe mice were exposed to a sublethal dose of LPS (2 mg/kg) i.p. to induce sepsis (endotoxemia) at 15 months of age[20] (Fig. 1a). The murine sepsis score (MSS)[21] was used to assess the severity and clinical status of sepsis, which included spontaneous activity, response to touch and auditory stimuli, posture, respiration rate and quality (labored breathing or gasping), and appearance (i.e., degree of piloerection) (Supplementary Table 1) at 0, 2, 4, 6, 12, 24,

48, and 72 h after LPS injection. LPS exposure induced sepsis in both Sed and Exe mice with a progressive increase of MSS, and decrease of body weight, body temperature, and blood glucose during the first 24 h after LPS injection. Compared to Sed mice, mice subjected to early-life exercise exhibited reduced sepsis severity, as demonstrated by a lower MSS at 12, 24, and 48 h post-LPS exposure (Fig. 1b), as well as an enhanced recovery from sepsis, as evidenced by a more rapid return to normal body weight (Fig. 1c), faster resolution of hypothermia (Fig. 1d) and hypoglycemia (Fig. 1e) between 24–72 h after LPS exposure. In addition, Exe mice exhibited significantly higher numbers of WBC, lymphocytes and monocytes in the blood at 6 h after LPS exposure compared with Sed mice (Fig. 1f–i). Exe mice also displayed reduced systemic inflammation and tissue inflammatory injury at 48 h after LPS exposure, as evidenced by decreased serum TNF and IL-1β levels (Fig. 1j, k), increased serum IL-1Ra level (Fig. 1l), reduced infiltration of inflammatory cells in the liver (Fig. 1m, n), less fat accumulation and vacuolization in the liver (Fig. 1o), and less inflammatory cell infiltration and interstitial edema in the lung (Supplementary Fig. 2).

In a separate set of experiments, both Sed and Exe mice were subjected to LPS challenge at the age of 4 months (Supplementary Fig. 3a). Consistent with previous findings, exercise training significantly ameliorated the severity of sepsis and promoted recovery from sepsis (Supplementary Fig. 3b–e). Moreover, exercise training increased WBC and lymphocytes numbers at 6 h after LPS exposure (Supplementary Fig. 3f–i), reduced serum levels of the pro-inflammatory cytokines including TNF, IL-1β and TIMP-1, increased the anti-inflammatory cytokine IL-1Ra (Supplementary Fig. 3j–m), mitigated liver, lung and spleen injury caused by LPS challenge (Supplementary Fig. 3n–p). Taken together, these data demonstrated that exercise during the early stages of life (1 to 4 months old) in mice enhanced anti-inflammatory immunity, as evidenced by restrained cytokine response triggered by LPS, reduced tissue damage and improved recovery from sepsis. Importantly, this enduring immunoregulating benefit extended into middle age (15 months old), indicating the potential long-term benefits of early-life exercise in improving anti-inflammatory immunity and mitigating the severity of infection later in life.

### Early-life exercise training induces a sustained elevation of circulating pipecolic acid in middle-aged mice

To elucidate the metabolic alterations resulting from exercise during early life, we conducted non-targeted steady-state metabolomics profiling on the serum and the liver of Exe and Sed mice at both 4 and 15 months of age. Orthogonal projection to latent structures-discriminant analysis (OPLS-DA) of serum metabolite profiles demonstrated significant differences between Exe and age-matched Sed mice at both time points (Fig. 2a, b). Of note, pipecolic acid was the only metabolite that was consistently upregulated in the serum of Exe mice compared to Sed mice at both 4 months (Fig. 2c and Supplementary Table 2) and 15 months (Fig. 2d and Supplementary Table 3) of age. Given that pipecolic acid is a product of lysine metabolism and that the liver is the primary site for lysine catabolism in mammals[22], we then analyzed the liver metabolite profile of Exe and Sed mice at 15 months of age (Fig. 2e and Supplementary Table 4). Pipecolic acid was among the top 20 differential metabolites in the variable importance in projection (VIP) score plot (considering $p$-value < 0.05 and VIP ≥ 1 as significant criteria), with a higher concentration in Exe mice than in Sed mice (Fig. 2f and Supplementary Table 4). Targeted LC-MS analysis validated that early-life exercise induced a significant increase of pipecolic acid in both the circulation (Fig. 2g) and the liver (Fig. 2h) of mice at 4 and 15 months of age, respectively. These data suggest that early-life exercise training, even after a long period of detraining, is correlated with upregulation of lysine degradation pathway and increased production of pipecolic acid in the liver.

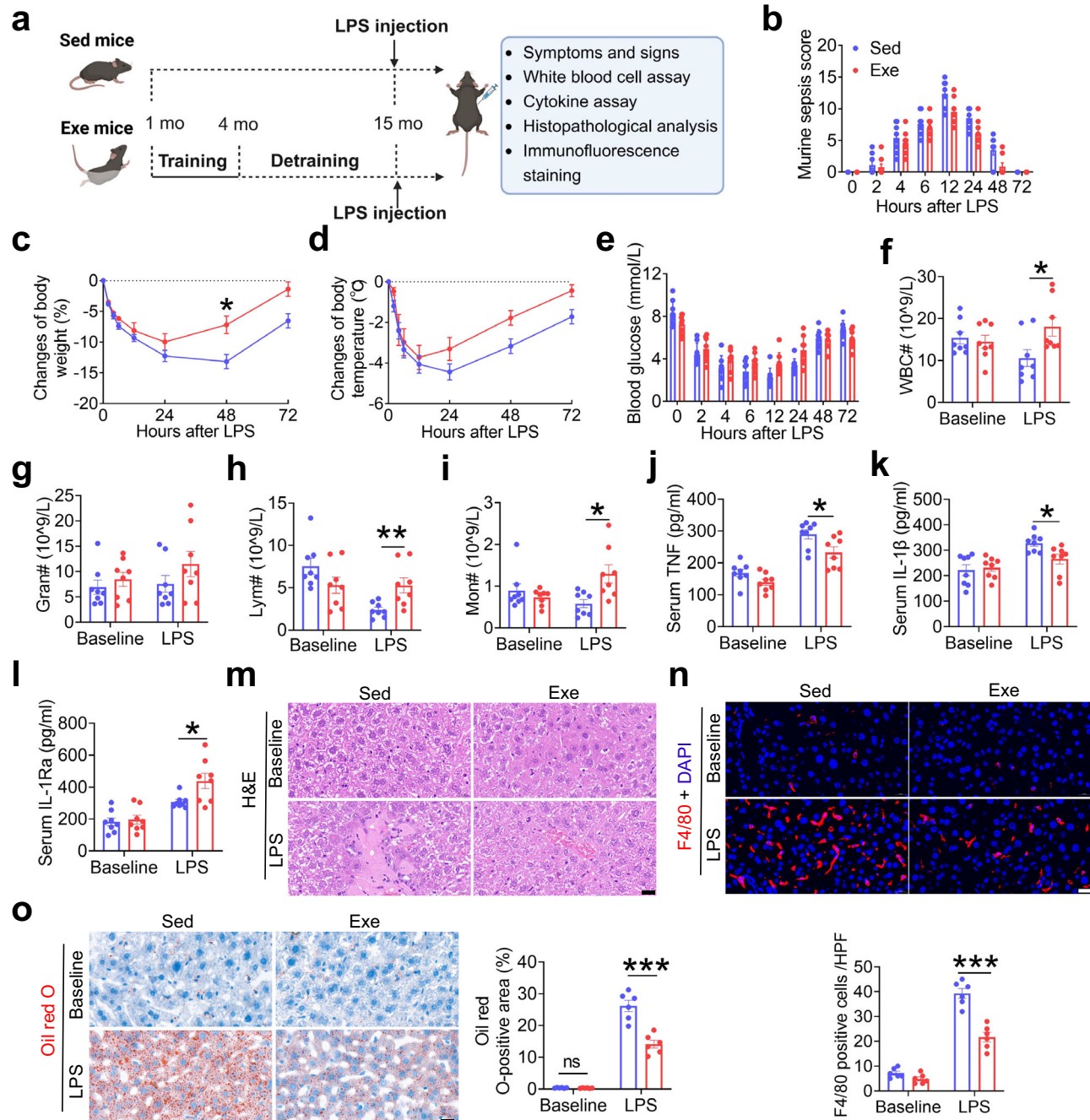

**Fig. 1 | Early-life exercise alleviates LPS-induced sepsis and promotes recovery in mice at middle-age. a** Experimental outline. One-month-old male mice were subjected to swimming training for 3 months, followed by 11 months of detraining period (Exe mice). Sed mice were keep detrained. All the mice were subjected to a single i.p. injection of ultrapure Escherichia coli O111:B4 strain LPS at 15 months old (created with BioRender.com). **b−e** Murine sepsis score (**b**), changes of body weight (**c**), changes of body temperature (**d**) and blood glucose levels (**E**) of mice at different time points (0, 2, 4, 6, 12, 24, 48 and 72 h after LPS injection). $n = 8$ per group. **f−i** Quantity of WBC, granulocyte, lymphocytes and monocytes in the blood at baseline and 6 h after LPS challenge. $n = 8$ per group. **j−l** Serum TNF, IL-1β and IL-1Ra levels at baseline and 48 h after LPS challenge. $n = 8$ per group. **m** Hematoxylin and eosin (H&E)-stained liver sections of mice at baseline and 48 h after LPS challenge.

Scale bar, 20 μm. **n** Representative images of F4/80 (red) stained liver sections of mice and the numbers of F4/80-positive cells in each high-power field (HPF) at baseline and 48 h after LPS challenge. $n = 6$ per group. Scale bar, 20 μm. **o** Representative images of Oil red O-stained liver sections of mice and the lipid droplet area of the liver sections at baseline and 48 h after LPS challenge. $n = 6$ per group. Scale bar, 20 μm. Values are presented as mean ± SEM. Data are analyzed by using the two-way repeated measures ANOVA followed by Bonferroni's test (**b−e**) or unpaired, two-tailed Student's t-test or Mann-Whitney U test (**f−o**). *$p < 0.05$, **$p < 0.01$, ***$p < 0.001$; exact $p$ values are listed in Source Data file. Sed=sedentary; Exe=exercise; WBC=white blood cell; Gran=granulocyte; Lym=lymphocyte; Mon=monocyte. Source data are provided as a Source Data file.

In order to explore the potential relevance of pipecolic acid to exercise in human, the circulating level of pipecolic acid was measured in healthy young volunteers (22.85 ± 3.40 years old) following a single bout exercise and in young highly-trained athletes (aged 16.21 ± 2.80 years). A single bout exercise, described specifically in our previous study[13], was found to induce a significant increase in circulating pipecolic acid in healthy young volunteers (Fig. 2i). Additionally, the serum level of pipecolic acid in young highly-trained athletes was significantly higher than that in inactive healthy individuals (trained vs. untrained, Fig. 2j). These findings showed that either acute or chronic exercise

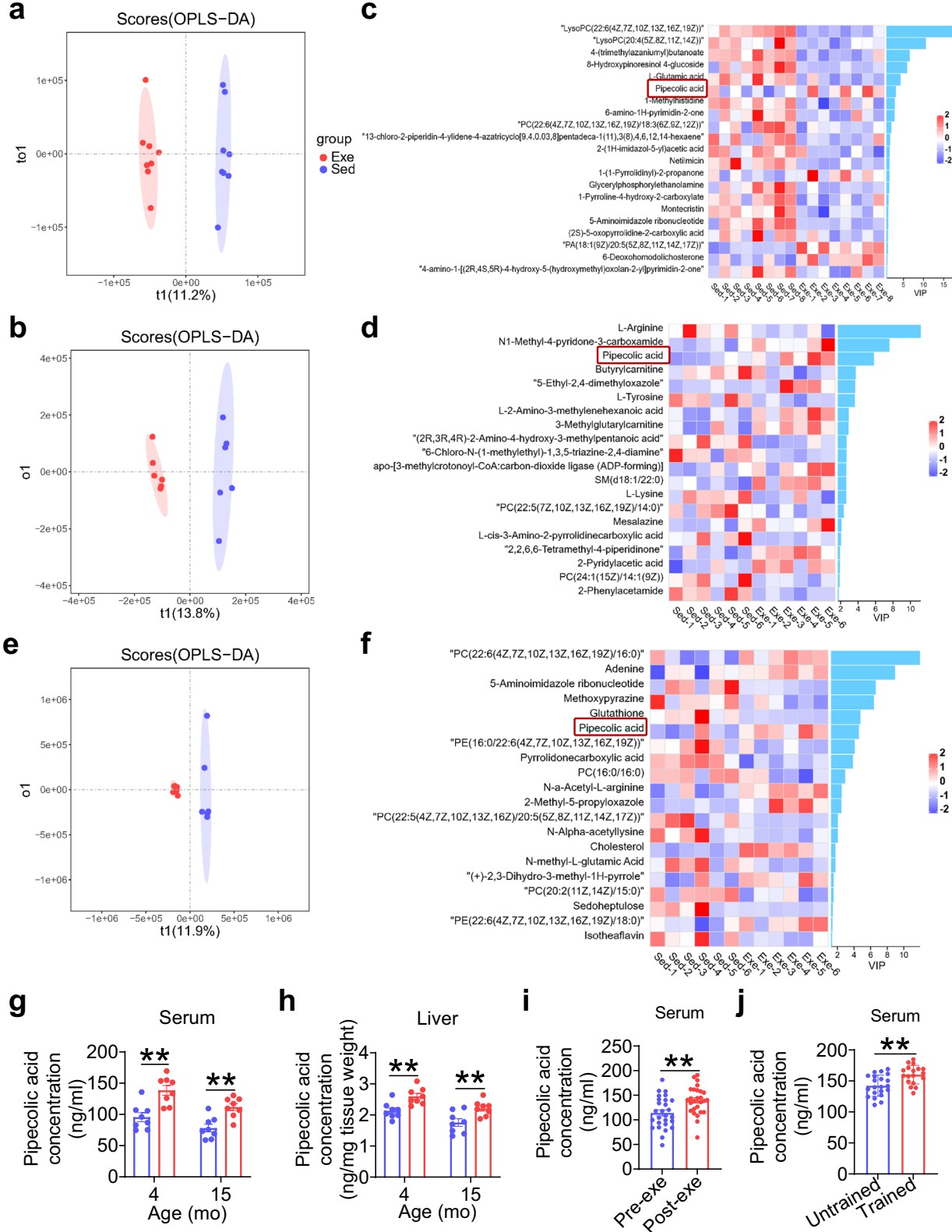

induces the elevation of circulating pipecolic acid, suggesting the role of pipecolic acid in exercise-induced metabolic changes in human.

### Pipecolic acid administration relieves LPS-induced sepsis and promotes recovery in mice

To investigate the potential therapeutic effects of pipecolic acid against LPS challenge, a 200 µg/kg body weight dose of pipecolic acid was administered intraperitoneally one hour prior to and 24 h following LPS injection (Fig. 3a); the effectiveness of pipecolic acid administration was validated by targeted LC-MS analysis that showed a substantial elevation in the circulating levels compared to vehicle-treated mice (Fig. 3b). At 6 h after LPS exposure, pipecolic acid administration significantly increased the numbers of WBC, granulocytes and monocytes in the blood (Fig. 3c–f). At 12, 24 and 48 h

**Fig. 2 | Early-life exercise training induces a sustained elevation of circulating pipecolic acid in mice. a–b** The clustering of orthogonal projection to latent structures-discriminant analysis (OPLS-DA) in positive ion mode of metabolite profile in serum of Sed and Exe mice at 4 months old (**a**) and 15 months old (**b**). **c–d** Variable Importance in Projection (VIP) score plot of the top 20 differential metabolites in serum of Sed and Exe mice at 4 months old (**c**) and 15 months old (**d**). $n = 6$–8 per group. **e** The OPLS-DA in positive ion mode of metabolite profile in liver. **f** VIP score plot of the top 20 differential metabolites in liver. $n = 6$ per group. **g–h** The concentration of pipecolic acid in serum and liver of Sed and Exe mice at 4 months old and 15 months old. $n = 8$ per group. **i** The pipecolic acid concentration in the serum of young healthy human volunteers after *vs.* before a single bout exercise. $n = 27$ per group. **j** The pipecolic acid concentration in the serum of young highly-trained athletes (trained group) *vs.* inactive healthy individuals (untrained group). $n = 21/18$ for untrained/trained participants. Values are presented as mean ± SEM. Data are analyzed by two-tailed unpaired Student's t test. **$p < 0.01$; exact $p$ values are listed in Source Data file. Sed=sedentary; Exe=exercise. Source data are provided as a Source Data file.

post-LPS injection, pipecolic acid-treated mice showed reduced MSS (Fig. 3g), and accelerated recovery from sepsis, as demonstrated by a quicker return of hypoglycemia (Fig. 3h), body weight (Fig. 3i) and hypothermia (Fig. 3j) within 72 h after LPS exposure. Moreover, pipecolic acid treatment significantly reduced serum TNF levels and increased IL-1Ra levels at 24 h after LPS exposure (Fig. 3k–m). Finally, pipecolic acid administration significantly reduced the infiltration of inflammatory cells, fat accumulation and vacuolization in the liver (Fig. 3n–p), and decreased inflammatory cell infiltration and interstitial edema in the lung of mice (Supplementary Fig. 4). These findings suggest that pipecolic acid administration ameliorates LPS-triggered inflammatory response, alleviates sepsis and facilitates recovery in mice.

## Pipecolic acid attenuates LPS-induced cytokine response in macrophages via inhibiting mTORC1 signaling

Macrophages play a crucial role in the pathogenesis of LPS-induced injury and the innate immune response. In order to explore the mechanisms underlying the anti-inflammatory effects of pipecolic acid in sepsis, bone marrow-derived macrophages (BMDMs) were isolated and treated with LPS in the presence of different concentrations of pipecolic acid (5, 10, 20 μM). Following a six-hour exposure to LPS (100 ng/ml), macrophages demonstrated significantly elevated mRNA levels of pro-inflammatory cytokines including TNF and IL-1β as well as inducible nitric oxide synthase (iNOS). However, treatment with pipecolic acid significantly reduced these mRNA levels in a dose-dependent manner (Fig. 4a–c). Subsequently, RNA-sequencing was conducted in BMDMs treated with LPS with or without pipecolic acid. The heatmap revealed distinct clusters for multiple samples (Supplementary Fig. 5a), while the Venn diagram indicated that 324 differentially expressed genes (DEGs) were upregulated in BMDMs following LPS stimulation but downregulated after pipecolic acid treatment (Supplementary Fig. 5b). Furthermore, KEGG analysis demonstrated that the mTOR signaling pathway was enriched among the 324 DEGs (Supplementary Fig. 5c), and GSEA analysis revealed that the mTORC1 pathway was activated by LPS stimulation but inhibited by pipecolic acid treatment (Supplementary Fig. 5d). Western blot analyzes confirmed that pipecolic acid inhibited mTORC1 activation in BMDMs exposed to LPS, as evidenced by significant reduction in the phosphorylation of phospho-p70 Ribosomal Protein S6 Kinase (p-p70 S6K, Thr389) and phospho-S6 Ribosomal Protein (p-S6, Ser240/244), which are downstream targets of mTOR (Fig. 4d). Furthermore, the combination of mTORC1 inhibitor rapamycin and pipecolic acid did not exert further inhibitory effect on pro-inflammatory cytokines and iNOS in LPS-stimulated macrophages (Fig. 4e–g). These findings suggest that pipecolic acid exerts an anti-inflammatory effect by inhibiting mTORC1 signaling in macrophages responding to LPS.

## The liver enzyme CRYM/ketimine reductase plays a crucial role in the long-term benefits of early-life exercise against LPS injury

Pipecolic acid is a metabolic product of lysine, primarily catalyzed by CRYM/ketimine reductase in the liver (Fig. 5a)[23]. Of the enzymes involved in the production and degradation of pipecolic acid (*Crym, Pycr1, Pipox, Aass, Hykk, Phykpl, Slc25a29*), only *Crym* mRNA

significantly increased in the liver of early-life Exe mice compared to Sed controls (Fig. 5b). Crym protein expression was also significantly higher in the liver of Exe mice compared to Sed controls (Fig. 5c). To further investigate the correlation between Crym and the benefits afforded by early-life exercise, 14-month-old mice (1-month-old mice trained with 3 months of swimming exercise followed by 10 months of detraining) were injected with adeno-associated virus 8 (AAV8) carrying either Crym-shRNA or the control scramble-shRNA (Fig. 5d). The AAV8 vector indicator mCherry fluorescence was enriched in the liver but not in the lung, heart, kidney or spleen of mice (Supplementary Fig. 6a). Both Crym-shRNA and scramble-shRNA AAV8 injection caused strong mCherry fluorescence in the liver as imaged by confocal microscopy (Supplementary Fig. 6b). As expected, Crym protein expression in the liver was significantly inhibited by Crym-shRNA but not scramble-shRNA (Fig. 5e). Pipecolic acid concentration either in the serum or in the liver was higher in Exe mice than Sed mice when they are injected with scramble-shRNA. This elevation of pipecolic acid caused by exercise training was blocked by injection of Crym-shRNA (Fig. 5f, g). In addition, scramble-shRNA-injected Exe mice displayed an increased number of WBCs, lymphocytes, and monocytes in the blood after LPS (Fig. 5h–k), a promoted recovery from sepsis (Fig. 5l–n), an alleviated systemic inflammation (Fig. 5o–q) and reduced tissue inflammatory injury (Fig. 5r–t, Supplementary Fig. 6c) compared to scramble-shRNA-injected Sed mice. In contrast, inhibition of Crym in the liver abolished the benefits of early-life exercise against LPS, as there was no significant difference in these indexes between Crym-shRNA-injected Sed mice and Exe mice (Fig. 5h–t, Supplementary Fig. 6c). These in vivo data indicate the essential role of Crym in the liver for persistent elevation of pipecolic acid and its contribution to the long-term benefits exerted by early-life exercise against LPS injury.

## Early-life exercise epigenetically promotes Crym expression in the liver

To unravel the mechanisms underlying the sustained higher expression of Crym in the liver induced by early-life exercise, we first investigated the DNA methylation status of *Crym* considering the well-established role of DNA methylation in modulating gene expression. However, we found no significant changes of DNA methylation between Exe mice and Sed mice within the CpG island of the *Crym* promoter region at the age of 15 months (Supplementary Fig. 7). We then explored histone epigenetic modifications and observed a significant increase of histone 3 lysine 4 trimethylation (H3K4me3) expression in the liver of 15-month-old mice exposed to early-life exercise (Fig. 6a). While H3K27me3 was also affected by exercise, their potency was not comparable to H3K4me3 (Fig. 6a). H3K4me3 is a well-established marker of transcriptional activation[24–26], and we thus conducted chromatin immunoprecipitation (ChIP) experiments on primary hepatocytes isolated from Exe mice and Sed mice both aged at 15 months. The ChIP-seq analysis revealed H3K4me3 occupancy at *Crym* promoter has a notable increase in hepatocytes of Exe mice (Fig. 6b). To verify this regulation, we designed primers that amplified different regions of *Crym* promoter, and found that H3K4me3 occupancy at −366 ~ −190 bp of *Crym* promoter was increased in hepatocytes of Exe mice (Fig. 6c). Consistently, H3K4me3 occupancy at this same region of *Crym* promoter was also increased in hepatocytes of

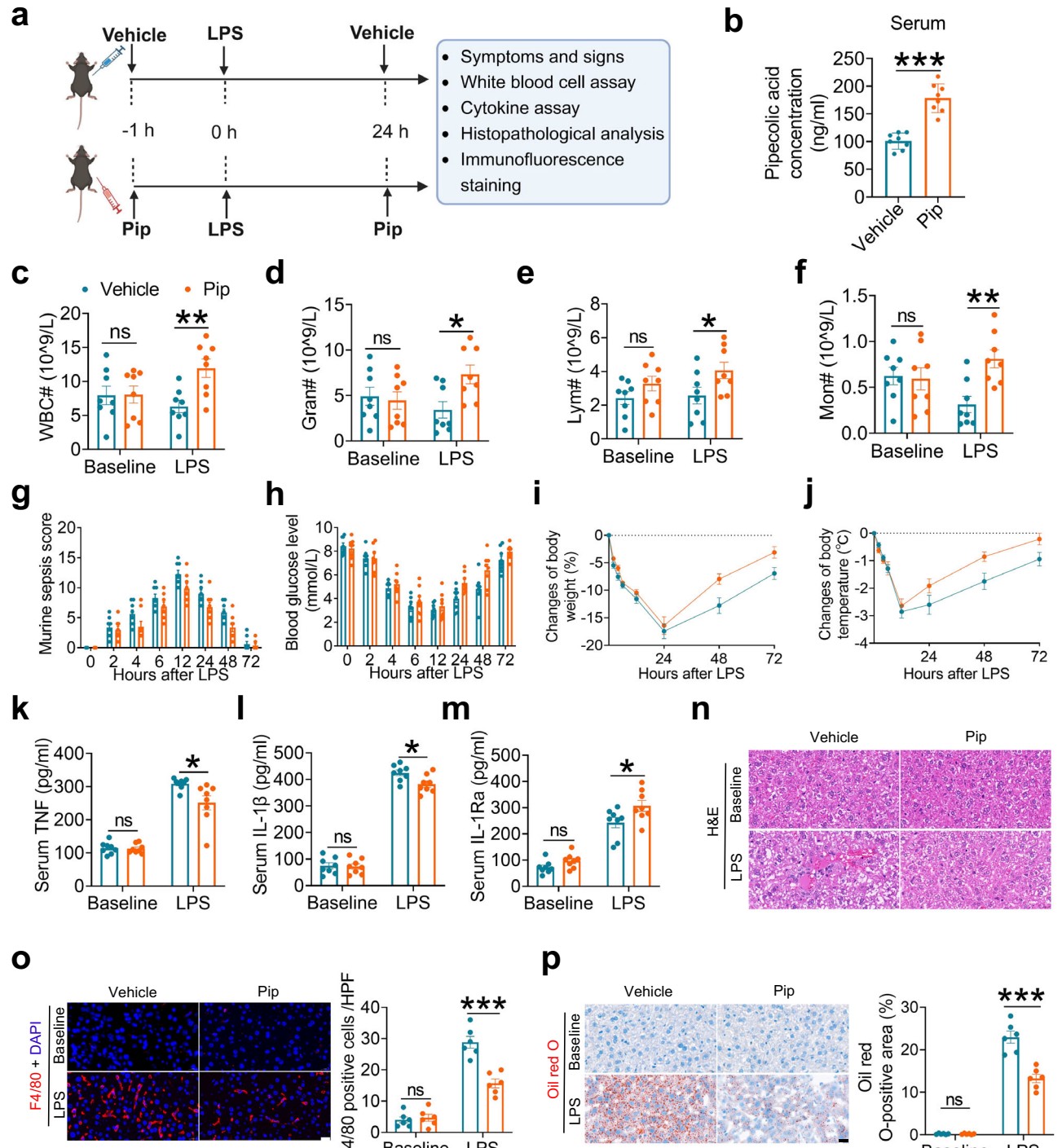

**Fig. 3 | Pipecolic acid administration relieves LPS-induced sepsis and promotes recovery in mice. a** Experimental outline. Six-week-old male mice were subjected to intraperitoneal injection with saline or LPS (2 mg/kg body weight, 1 h) with or without intraperitoneal pipecolic acid (200 μg/kg) pre-treatment (0 h) and treatment (24 h) (created with BioRender.com). **b** The concentration of pipecolic acid in serum of saline and pipecolic acid pretreated mice immediately before LPS injection. $n = 8$ per group. **c–f** Quantity of WBCs, granulocytes, lymphocytes and monocytes in the blood at baseline and 6 h after LPS challenge. $n = 8$ per group. **g–j** Clinical severity score (**g**), blood glucose level (**h**), changes of body weight (**i**), changes of body temperature (**j**) of mice at different time points (0, 2, 4, 6, 12, 24, 48 and 72 h after LPS injection). $n = 8$ per group. **k–m** Serum TNF, IL-1β and IL-1Ra levels at baseline and 24 h after LPS challenge. $n = 8$ per group. **n** Representative

images of hematoxylin and eosin (H&E)-stained liver sections of mice at baseline and 48 h after LPS challenge. Scale bar, 20 μm. **o** Representative images of F4/80 (red) stained liver sections of mice, and the numbers of F4/80-positive cells in each high-power field (HPF) at baseline and 48 h after LPS challenge. $n = 6$ per group. Scale bar, 20 μm. **p** Representative images of Oil red O-stained liver sections of mice, and the lipid droplet area of the liver sections at baseline and 48 h after LPS challenge. $n = 6$ per group. Scale bar, 20 μm. Values are presented as mean ± SEM. Data are analyzed using unpaired, two-tailed Student's t test or Mann-Whitney U test (**b–f**, **k–p**) and two-way repeated measures ANOVA followed by Bonferroni's test (**g-j**). ns, no significance; *$p < 0.05$, **$p < 0.01$, ***$p < 0.001$; exact p values are listed in Source Data file. Pip=pipecolic acid; WBC=white blood cell; Gran=granulocyte; Lym=lymphocyte; Mon=monocyte.

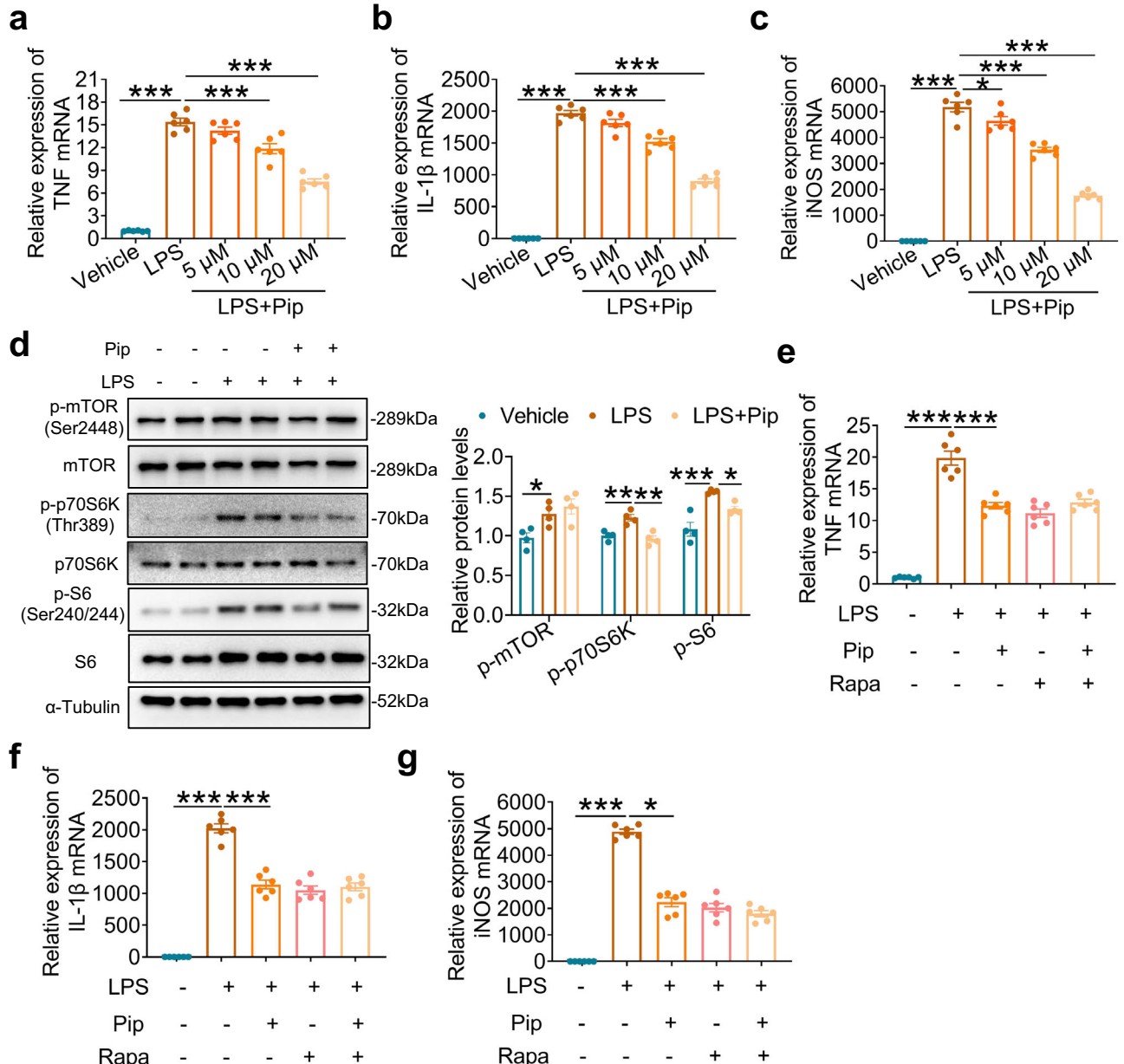

**Fig. 4 | Pipecolic acid attenuates LPS-induced cytokine response in macrophages via inhibiting mTORC1 signaling. a–c** qRT-PCR determination of pro-inflammatory cytokines (TNF, IL-1β, and iNOS) mRNA. BMDMs were stimulated with different doses of pipecolic acid (5, 10, 20 μM) and 100 ng/ml LPS for 6 h. $n = 6$ per group. **d** Representative blots and quantified data of the expression of phospho-mTOR (Ser2448), mTOR, phosphop-p70s6k (Thr389), p70s6k, phosphop-S6 (Ser240/244), S6 in the BMDMs. BMDMs were pre-treated with 20 μM pipecolic acid for 1 h before LPS (100 ng/ml, 6 h) stimulation. $n = 4$ per group. **e–g** qRT-PCR determination of pro-inflammatory cytokines (TNF, IL-1β, and iNOS) mRNA. BMDMs were pre-treated with 100 nM rapamycin for 1 h before pipecolic acid (20 μM, 6 h) and LPS (100 ng/ml, 6 h) stimulation. $n = 6$ per group. Values are presented as mean ± SEM. Data are analyzed using one-way ANOVA followed by Bonferroni's test. *$p < 0.05$; **$p < 0.01$; ***$p < 0.001$; exact $p$ values are listed in Source Data file. Pip=pipecolic acid; Rapa=rapamycin. Source data are provided as a Source Data file.

4-month-old Exe mice compared with Sed mice (Fig. 6d). To identify which H3K4 methyltransferases contributed the altered H3K4me3 modification, we performed mRNA analysis of *Kmt2a, Kmt2b, Kmt2c, Kmt2d, Kmt2e, Setd1a,* and *Setd1b.* Among these H3K4 methyltransferases, *Setd1a* is the only one that significantly increased in the hepatocytes of Exe mice compared to Sed mice both at the age of 4 months and 15 months (Fig. 6e, f). To further investigate the regulatory role of SETD1A in Crym expression, mouse primary hepatocytes were transfected with SETD1A-overexpressing adenovirus (Ad-SETD1A) or a control Ad-eGFP. Western blot and qPCR data confirmed

SETD1A overexpression in Ad-SETD1A-transfected hepatocytes (Fig. 6g–i). SETD1A overexpression led to increased H3K4me3 occupancy at *Crym* promoter (Fig. 6j), elevation of Crym mRNA and protein expressions (Fig. 6g–i) in hepatocytes. Conversely, SETD1A knockdown by siRNA (Fig. 6k, l) decreased H3K4me3 occupancy at *Crym* promoter (Fig. 6m), reduced Crym mRNA and protein expressions in hepatocytes (Fig. 6k, l). Taken together, these findings have indicated that SETD1A-mediated H3K4me3 epigenomic modification in the *Crym* promoter in hepatocytes contributes to the elevation of Crym expression induced by early-life exercise.

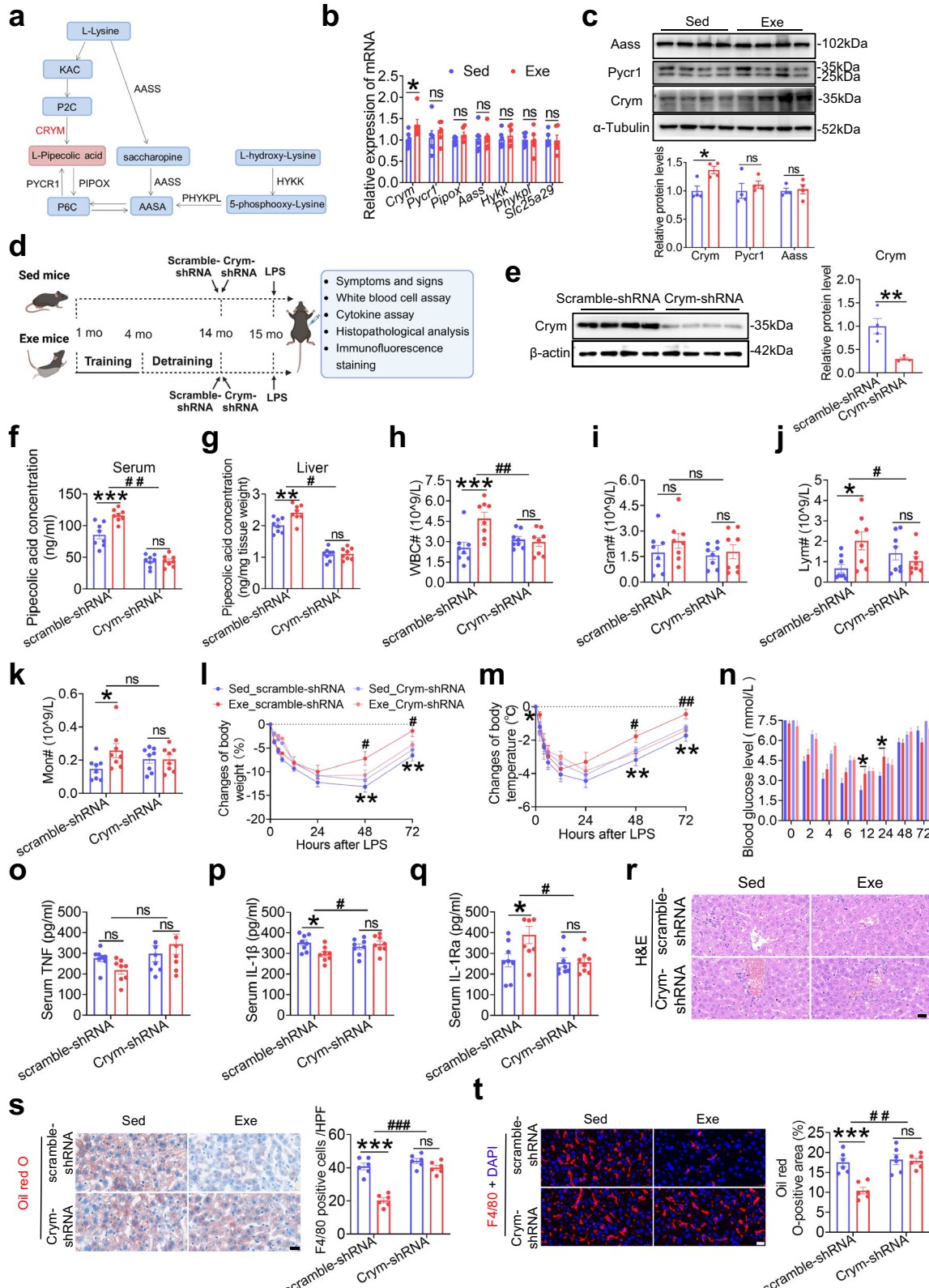

## Discussion

This study has made significant contributions to our understanding of the long-term effects of exercise on immune health through epigenetic modulation of immunometabolic network signaling. First, we demonstrated that early-life exercise training curbed inflammatory response to LPS challenge and mitigated sepsis in middle-aged male mice, even after a long period of detraining. Second, this persistent immunoregulating effect of early-life exercise is mainly attributed to increased production of pipecolic acid, which is related to increased H3K4me3 at the *Crym* promoter. The potential relevance of pipecolic acid to exercise is also evidenced by elevation of circulating pipecolic acid in healthy young volunteers following acute exercise and in young highly-trained athletes. Thirdly, pipecolic acid exerts a significant anti-inflammatory effect via reducing pro-inflammatory cytokines in

**Fig. 5 | The liver enzyme CRYM/ketimine reductase plays a crucial role in the long-term benefits of early-life exercise against LPS injury. a** The lysine degradation pathway. **b** Relative mRNA level of the enzymes involved in the production and degradation of pipecolic acid in the liver of Sed and Exe mice at age of 15 months. $n = 8$ per group. **c** Western blot and its quantitative results of enzymes in the liver of Sed and Exe mice at age of 15 months. $n = 4$ per group. **d** Sed and Exe mice were injected with AAV8 either carrying shRNA targeting Crym (Crym-shRNA) or scramble control (scramble-shRNA) aged at 14 months (created with BioRender.com). **e** Representative blots and quantification of Crym protein expression in the liver. $n = 4$ per group. **f–g** Serum and liver pipecolic acid level in the Sed and Exe mice aged at 15 months. $n = 8$ per group. **h–k** Quantification of WBC, granulocyte, lymphocytes and monocytes in blood 6 h after LPS challenge. $n = 8$ per group. **l–n** Body weight (**l**), body temperature (**m**), blood glucose levels (**n**) of mice at different time points after LPS injection. $n = 8$ per group. **o–q** Serum TNF, IL-1β and IL-1Ra levels at 48 h after LPS challenge. $n = 8$ per group. **r–t** Representative images of hematoxylin and eosin (H&E)-stained liver sections of mice (**r**), F4/80 (red) stained liver sections of mice and the numbers of F4/80-positive cells (**s**), and Oil red O-stained liver sections of mice and the lipid droplet area of the liver sections (**t**) at 48 h after LPS challenge. $n = 6$ per group. Scale bar, 20 μm. Values are presented as mean ± SEM. Data are analyzed using two-tailed Student's t-test or Mann-Whitney U test (**b–c, e**) or two-way ANOVA followed by Bonferroni's test (**f–k, o–t, l–n** at each time point). ns, no significance; *$p < 0.05$; **$p < 0.01$, ***$p < 0.001$ with the comparisons indicated by the lines; #$p < 0.05$, ##$p < 0.01$ for interaction using two-way ANOVA; exact $p$ values are listed in Source Data file. Source data are provided as a Source Data file.

macrophages through inhibiting mTORC1 signaling, alleviating LPS-induced sepsis injury and facilitating recovery in mice (Fig. 7).

Repeated bouts of moderate exercise have been shown to boost immunosurveillance and offer multiple health benefits[27]. Previous randomized clinical trials have demonstrated that moderate exercise programs help protect against acute respiratory infection[28,29]. Consistently meeting physical activity guidelines is strongly associated with a reduced risk for severe COVID-19 outcomes among infected adults[30]. Animal studies reveal that a 4-week aerobic exercise pretreatment improved LPS-induced acute lung injury in mice[31,32]. In the present study, exercise during the early stage of life (1–4 months old) in mice alleviated LPS-induced sepsis and improved recovery. Notably, this enduring immune benefit extended to middle-aged phase (15 months old), enriching the current understanding of long-term health benefits of exercise.

Frodermann et al.'s study[33] showed that mice with access to running wheels, on which they ran voluntarily for over 12 km each night, experienced a reduction in all leukocytes due to decreased hematopoietic progenitor cell proliferation in their bone marrow. However, exercise did not inhibit emergency hematopoiesis during infection. This was evidenced by hematopoietic progenitors of exercising mice responding vigorously, with 1.4-fold increase of c.f.u compared to progenitors from sedentary littermates when exposed to LPS-induced sepsis. In the present study, we had similar finding that mice subjected to 3-month swimming training had reduced WBC number but a significant increase of WBC number when exposed to LPS (Supplementary Fig. 3f). After 11 months of detraining, the WBC number returned to normal and showed no significant difference in exercised mice compared with sedentary mice. However, early-life exercised mice showed a significantly increased WBC number at 6 h after LPS exposure and reduced systemic inflammation at 48 h after LPS exposure, as evidenced by decreased serum TNF and IL-1β levels, increased serum IL-1Ra level, reduced infiltration of inflammatory cells in the liver and lung. reduced pro-inflammatory cytokines including compared with sedentary mice, indicating that the early-life exercise has sustained immunoregulating benefits. The baseline serum levels of IL-1β in both Exe and Sed mice in our experiments were higher than what were typically reported in the literature. Several factors may contribute to this variation, including differences in mouse strain, age, animal housing, experimental conditions, sampling technique, as well as variations in the brands and batches of assay kits used. A recent study by Tahtinen S et al.[34] reported that IL-1 and IL-1ra are key regulators of the inflammatory response to RNA vaccines and other types of innate immune stimulation including LPS. Their study demonstrated a markedly greater induction of IL-1ra compared to IL-1α or IL-1β, offering protection against uncontrolled systemic inflammation, while the "buffering capacity" of IL-1ra is likely to be overcome as the magnitude of innate stimulation is increased. In the present study, we observed comparable levels of serum IL-1Ra level and IL-1β, which might be attributed to relatively higher dose of LPS used in our study.

Exercise-induced metabolic responses, particularly those involving lipids and amino acids, play a crucial role in the complex interplay between physical activity and overall health[14]. Pipecolic acid is a metabolic product of lysine and the liver is the primary organ for lysine catabolism in mammals[22]. A cohort study identified that pipecolic acid may be a promising atheroprotective candidate for mediation of sports activity on atherosclerosis[35]. Another randomized, double-blind, placebo-controlled, primary prevention trial showed that pipecolic acid is positively associated with fruit and vegetable intake and a healthy diet, which can reflect the long-term impact of the overall diet[36]. In the present study, pipecolic acid attenuates pro-inflammatory cytokine production in LPS-induced macrophages and relieves LPS-induced sepsis in mice, indicating pipecolic acid exerts an anti-inflammatory effect. In addition, our data have shown that exercise training in early-life induced an elevation of pipecolic acid in the liver and serum, which persists until middle age. These findings suggest that pipecolic acid is a key intermediate metabolite mediating the long-term beneficial effects of exercise.

mTOR is a cellular energy sensor regulated by exercise and serves in the immune system to enhance inflammatory and effector functions[37]. Our previous data have reported the importance of mTOR signaling pathway in maintaining innate immunity homeostasis[38]. GSEA analysis of RNA-sequencing revealed that the mTORC1 pathway was activated by LPS stimulation but inhibited by pipecolic acid treatment in macrophages (Supplementary Fig. 5d). Further experiments confirmed that pipecolic acid inhibited mTORC1 activation in LPS-stimulated macrophages and the combination of rapamycin and pipecolic acid did not exert further inhibitory effect on cytokines production. These data have indicated that mTORC1 pathway plays a role in the anti-inflammatory effect of pipecolic acid in macrophages responding to LPS. Considering TNF signaling pathway was also enriched among the differentially expressed genes in KEGG analysis (Supplementary Fig. 5c), How pipecolic acid affects TNF signaling pathway is also an interesting issue warranting future studies.

This study has several limitations. First, the long-term effect of exercise was only investigated in a mouse model in the present study. While we observed an increase in the circulating level of pipecolic acid in young human volunteers following a single bout exercise, and found higher levels in young highly-trained athletes, the available data is limited to draw a conclusion regarding the long-lasting immune benefits of early-life exercise in humans. Further studies are warranted to investigate the potential correlation between pipecolic acid and immune benefits in athletes during middle-age and detrained periods. Second, the mouse sepsis model in this study was induced by LPS rather than a live pathogen. Since LPS is a single component released by gram-negative organisms, it neglects the host-pathogen interactions of gram-positive organisms and polymicrobial sepsis. Therefore, to comprehensively investigate the immunological benefits of exercise, a more suitable sepsis model, such as bacterial injection or cecal

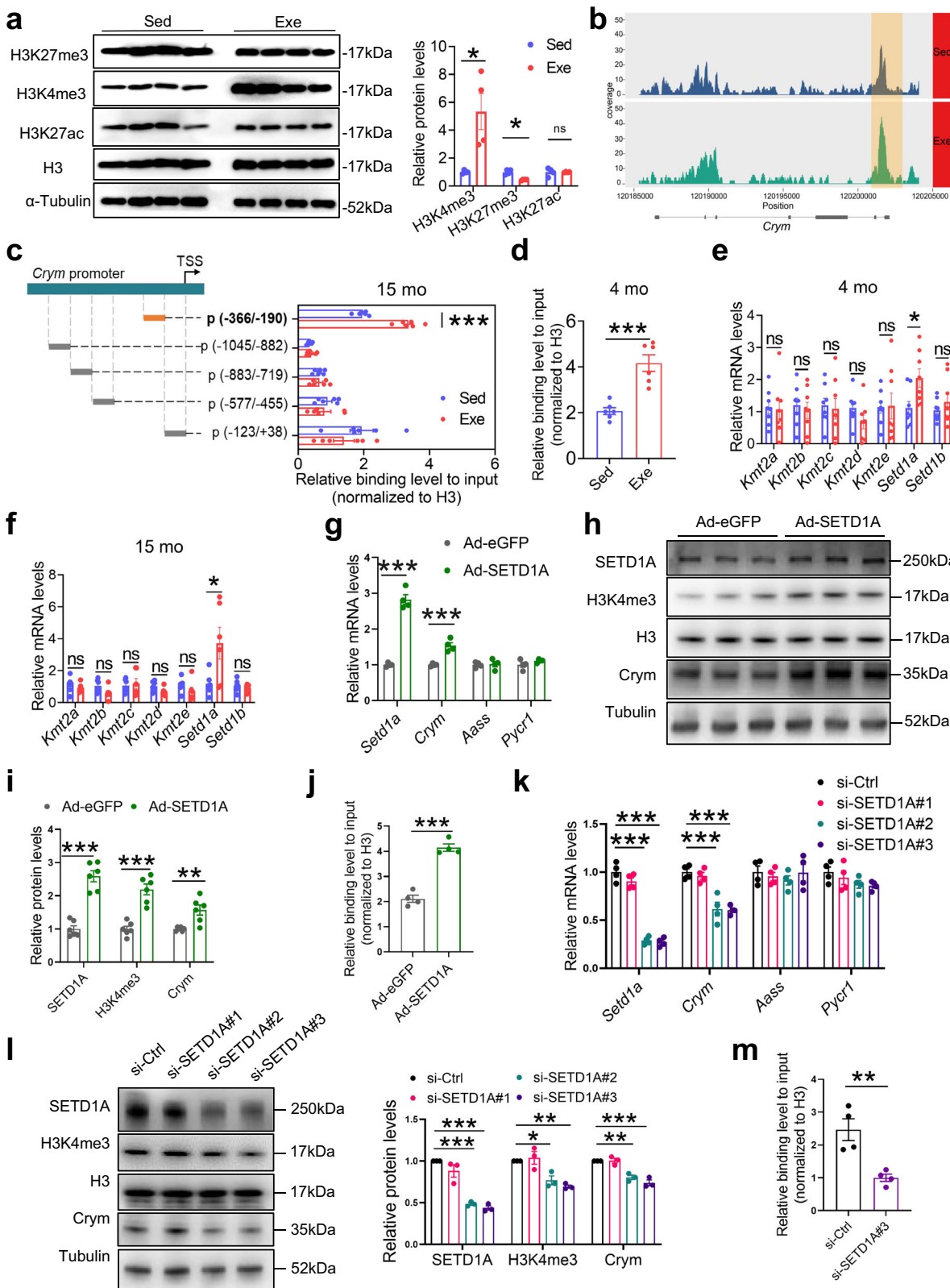

ligation and puncture is suggested to be used in future studies. Third, although our in vitro data demonstrated that Setd1a-mediated H3K4me3 epigenomic modification contributed greatly to the elevation of Crym expression in the liver, the causal role between H3K4me3 epigenomic modification and Crym expression needs to be further elucidated by studies on animals and humans.

In conclusion, our findings demonstrate that early-life regular exercise yields an enhanced anti-inflammatory immunity until middle-aged phase in mice. This enduring benefit is mediated by epigenetic modulation of immunometabolic network signaling, in which pipe-colic acid plays a pivotal role. This highlights the significance of early-life regular physical activity to immune health in later life.

**Fig. 6 | Early-life exercise epigenetically modulates *Crym* expression in the liver.**
**a** Representative blots and quantified H3K4me3, H3K27me3 and H3K27ac protein expression in the liver of Sed and Exe mice aged at 15 months. *n* = 4 per group. **b** Representative images showing ChIP-seq peaks against H3K4me3 on *Crym* promoter in the liver of Sed and Exe mice aged at 15 months. **c** Relative enrichment of H3K4me3 on the indicated regions of *Crym* promoter in primary hepatocytes isolated from Sed and Exe mice aged at 15 months. *n* = 6 per group. **d** Relative enrichment of H3K4me3 on p (-366/-190) of *Crym* promoter in primary hepatocytes isolated from Sed and Exe mice aged at 4 months. *n* = 6 per group. **e–f** Relative mRNA level of histone methylation-related genes in the hepatocytes of Exe and Sed mice aged at 4 months (**e**) and 15 months (**f**). *n* = 6–8 per group. **g–j** Relative mRNA expression of indicated genes (**g**), representative blots (**h**) and quantified

expression of indicated proteins (**i**), and relative enrichment of H3K4me3 on p (-366/−190) of *Crym* promoter (**j**) in mouse primary hepatocytes transfected with Ad-SEDT1A or Ad-eGFP. *n* = 4 per group in **g** and **j**, *n* = 6 per group in **i**. **k–m** Relative mRNA expressions (**k**), representative blots and quantified protein expressions (**l**), and relative enrichment of H3K4me3 on *Crym* promoter (**m**) in mouse primary hepatocytes transfected with si-SETD1A or si-Ctrl for 48 h. *n* = 4 per group. Values are presented as mean ± SEM. Data are analyzed by unpaired, two-tailed Student's t-test or Mann-Whitney U test (**a**, **c–j**, **m**) or one-way ANOVA followed by Bonferroni's test (**k**, **l**). ns, no significance; *$p < 0.05$; **$p < 0.01$; ***$p < 0.001$; exact p values are listed in Source Data file. Sed=sedentary; Exe=exercise. Source data are provided as a Source Data file.

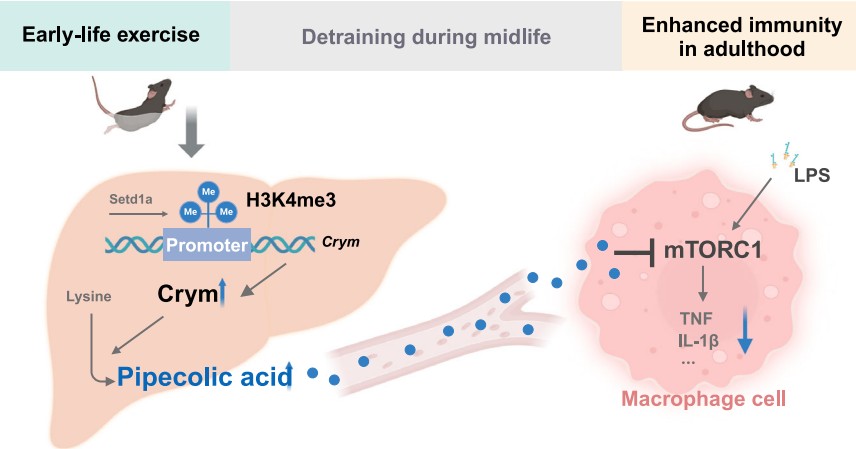

**Fig. 7 | Schematic depicting long-term effects of early-life exercise on immunity.** Early-life exercise training curbs inflammatory response to LPS challenge and mitigates sepsis in adulthood in mice, even after a long period of detraining. This persistent immunoregulatory effect is mainly attributed to enhanced hepatic production of pipecolic acid due to increased H3K4me3 modification at the Crym promoter. Pipecolic acid exerts a significant anti-inflammatory effect via reducing pro-inflammatory cytokines in macrophages through inhibiting mTORC1 signaling (created with BioRender.com).

## Methods

### Study approvals
All human studies were conducted according to the Declaration of Helsinki principles and were approved by the Human Research Ethics Committee of Xi'an Physical Education University (Approval No. XAIPE2023011). All animal experiments were approved by the Laboratory Animal Welfare and Ethics Committee of Fourth Military Medical University (Approval No. 20180303). The care and protection of experimental animals were in accordance with the Guide for the Care and Use of Laboratory Animals published by the US National Institutes of Health.

### Participants
All participants signed a consent form after reading information about the study and having the procedures explained to them. For those under 18 years old, the consent from their parents was obtained. A total of 39 healthy young (aged 16.21 ± 2.80 years) were included: 18 young highly-trained athletes (trained group) and 21 inactive healthy individuals (untrained group), all of whom are male (Supplementary Table 5). The young athletes had performed organized, team-based skiing training for over 1 year, whereas the inactive healthy individuals had no regular exercise for over 1 year and had abstained from exercise at least 1 week before the study. In another previous experiment from our lab[13], a total of 27 young healthy volunteers were recruited (aged 22.85 ± 3.40 years), of whom 18 adults are male and 9 adults are female. Each participant underwent a constant load submaximal exercise, which was started with 0 W, 60 round per min for 2 min as warm-up, then reached 100 W (male)/70 W (female) within 1 min and kept the intensity for 20 min with cadence of 60 round per min, and ended with a 2 min cool-down at room temperature (25 °C).

### Animals and the exercise protocol
One-month-old male C57BL/6 J mice were obtained from the Animal Center of Fourth Military Medical University. Animals were maintained in a 12-hour light/12-hour dark cycle and allowed access to water and diet (#DOSSYJY-001, DOSSY) ad libitum. Ambient temperature (22 °C) and 30–70% humidity was maintained. The swim training protocol was modified from our previously established procedure[39]. Mice (1 month) swam 60 min once daily for 5 days a week for 12 weeks. Mice were adapted to swimming training with a 10 min session on the first day. Sessions were then progressively increased to 60 min/day over a 1-week period. All exercise sessions took place during 16:30 p.m.–18:00 p.m. Age-matched sedentary control mice were allowed to swim for only 30 s each day and were then gently dried. Following three months of exercise training, mice were routinely housed without further training until they reached 15 months of age, and then subjected to LPS challenge.

### Sepsis model and clinical severity assessment
Sepsis induction was carried out in mice by a single i.p. injection of ultrapure *Escherichia coli* O55:B5 strain LPS (2 mg/kg body weight, dissolved in 0.1 ml/10 g body weight saline, S8776; Merck, Darmstadt, Hesse, Germany). At 0 (baseline), 2, 4, 6, 12, 24, 48, and 72 h post LPS injection, body weight, blood glucose levels, and body temperature were measured and the murine sepsis score (MSS) was evaluated blindly as described previously (Supplementary Table 1)[21]. Blood glucose levels were measured by blood glucose meter and respective test strips (ACCU-CHEK performa, Penzberg, Bayern, Germany).

Mice were euthanized by excessive inhalation of isoflurane and bled completely at 0, 24, 48 and 72 h after LPS injection. Blood was collected in a 1.5 ml tube for clotting. After 30 min at room

temperature, clotted blood was centrifuged at 1500 g for 10 min to obtain serum which was used for cytokine assay. Livers and lungs were removed and either fixed in 4% paraformaldehyde solution or snap-frozen in liquid nitrogen for biochemical analysis.

## Complete blood cell assay and cytokine assay

Blood samples were collected using a disposable anticoagulant tube containing EDTA and analyzed using an Element Ht5 Auto Hematology analyzer (Heska, Canada) for automatic Complete Blood Count within 6 h of collection. Blood samples were collected using disposable 1.5 ml centrifuge tubes for obtaining serum. Relative levels of cytokines and chemokines were determined by using the Mouse cytokine array panel A array kit (ARY006, R&D Systems, Minneapolis, USA). Selected cytokines, including TNF, IL-1β and IL-1RA were measured by ELISA kits (CSB-E04741, CSB-E08054 & CSB-E10395, CUSABIO, Wuhan, China).

## Histopathological analysis

Liver and lung tissues were either frozen quickly or fixed with 4% paraformaldehyde for more than 24 h. Hematoxylin and eosin (H&E) staining was performed in fixed samples and sections which were cut at a thickness of 5 μm. Lipid droplet accumulation in the liver was visualized using Oil red O (O0625; Merck, Darmstadt, Hesse, Germany) staining of frozen liver sections that were prepared in optimum cutting temperature (O.C.T.) compound (4583; Tissue-Tek, Sakura). Histopathology images were captured with a Nikon E200 light microscope (Nikon, Tokyo, Japan).

## Immunofluorescence staining

For immunofluorescence microscopy, paraffin sections were labeled with primary antibodies overnight, followed by incubation with a suitable fluorophore-conjugated secondary antibody for 1 h. Specifically, rat anti-mouse-F4/80 antibody (ab6640, 1:50; Abcam) were used to stain infiltrated macrophages and Kupffer cells in liver sections. Immunofluorescence images were obtained using an inverted confocal microscope (Zeiss LSM 800). Positive-staining cells in the images were quantified using Image-Pro Plus software (version 6.0, Media cybernetics).

## Untargeted Liquid Chromatography-Mass Spectrometry (LC-MS)

Blood samples were taken from 4-month-old mice with 3 months of swim training and 15-month-old mice with 3 months of swim training and 11 months of detraining. Liver samples were collected from 15-month-old mice with 3 months of swim training and 11 months of detraining. Metabolomic profiling was conducted by liquid chromatography tandem mass spectrometry (LC-MS) on the UHPLC system (Agilent Technologies, Santa Clara, CA, USA, #1290) by Gene Denovo Biotechnology Co. (Guangzhou, China). After data pre-processing and annotation, multivariate statistical analysis was performed. Orthogonal projection to latent structures-discriminant analysis (OPLS-DA) was used in comparison groups with R package models. A variable importance in projection (VIP) score was used to rank the metabolites distinguished between the two groups. A t-test was employed to screen differential metabolites, with a p-value < 0.05 and VIP ≥ 1 considered as significant criteria. The VIP score of OPLS-DA was used to create a graph. And the top 20 metabolites, ranked in descending order, were depicted in the VIP score plot.

## Pipecolic acid concentration analysis

Samples (20 μl serum or 50 mg tissue) were homogenized with methanol to extract the metabolites, followed by mixing with internal standard (P45850; Merk, Darmstadt, Hesse, Germany). Subsequently, the derivatized samples were subjected to ultra-performance liquid chromatography coupled to a tandem mass spectrometry (UPLC-MS/MS) system (ACQUITY UPLC-Xevo TQ-S, Waters Corp., MA, USA) to quantitate the metabolites. Raw data from UPLC-MS/MS analysis were analyzed using Mass-Lynx software (ver. 4.1, Waters Corp.) to calculate the concentration of pipecolic acid in the samples.

## qRT-PCR

Cellular or tissue RNAs were extracted using TRI reagent (T9424; Merck, Darmstadt, Hesse, Germany). Reverse transcription was performed using 500 ng of total RNA with the Evo M-MLV Reverse Transcription Kit (AG11706; Accurate Biology, Hunan, China). Reactions were performed in technical and biological triplicate using a 96-well format on an ABI Step One Plus system, using Hieff qPCR SYBR Green Kit (11201; YESEN Biology, Shanghai, China). The quantitative reverse transcription-polymerase chain reaction (qRT-PCR) conditions were 95 °C for 5 min, followed by 40 cycles of 95 °C for 10 s and 60 °C for 30 s. Gene expression was normalized to β-actin or 36B4, and relative expression was calculated using the $2^{-\Delta\Delta Ct}$ method. Primers were used at 0.5 μM, and their sequences are listed in Supplementary Table 6. PCR efficiency was optimized and melting curve analyses of products were performed to ensure reaction specificity.

## Western blot

The protein expression levels were measured by western blot as described previously[40]. Proteins from tissues and cells were isolated using radioimmunoprecipitation assay lysis buffer (P0013B; Beyotime, Shanghai, China) containing protease inhibitor cocktail, EDTA Free, 100×(APT006; AntiProtech, Texas, USA) and phosphatase inhibitor cocktail, 100× (APT008; AntiProtech, Texas, USA). The protein concentration was measured with a bicinchoninic acid protein assay kit (P0012; Beyotime, Shanghai, China). Western blot analysis was performed using standard procedures, was detected using the Pierce enhanced chemiluminescence western blotting detection kit (32106; Thermo Fisher, Waltham, MA, USA), and was quantified by scanning densitometry. Primary antibodies against phospho-mTOR (Ser2448) (1:1000, 5536), mTOR (1:1000, 2983), phospho-p70S6K (Thr389) (1:1000, 9234), total p70S6K (1:1000, 2708), phospho-S6 ribosomal protein (Ser240/244) (1:1000, 2215), total S6 ribosomal protein (1:1000, 2217), H3K4me3 (1:1000, 9751), H3K27me3 (1:1000, 9733), H3K27ac (1:1000, 8173), histone-H3 (1:2000, 4499), α-Tubulin (1:1000, 2125) and β-actin (1:1000, 4970) were purchased from Cell Signaling Technology (Danvers, MA, USA). Aass (1:2000, sc-365417) and Crym (1:500, sc-376687) were purchased from Santa Cruz Biotechnology (Dallas, Texas, USA). Pycr1 (1:500, ab279385) was purchased from Abcam (Cambridge, UK). SETD1A (1:1000, 67936-1-lg) was purchased from Proteintech (Wuhan, China). α-Tubulin or β-actin was used as a loading control. Unprocessed blots are supplied in the source data file or at the end of the Supplementary information.

## Isolation and differentiation of bone marrow-derived macrophages (BMDMs)

Mice were euthanized by excessive inhalation of isoflurane. Bone marrow cavities of the femur and tibia of mice were flushed with 5 ml cold, sterile PBS (02-020-1 A; Biological Industries, Israel). After lysing red blood cells, the bone marrow cells were washed, resuspended, and differentiated into macrophages in DMEM (SH30021.01B; HyClone, Logan, Utah, USA) with 10% FBS (SH30070.03; HyClone, Logan, Utah, USA), 20 ng/ml recombinant murine M-CSF (SRP3221; Merck, Darmstadt, Hesse, Germany) conditioned medium, 1% penicillin/streptomycin, 100×(P400; Solarbio life science, Beijing, China). Cells were differently treated for further experiments after differentiation for 6–7 days.

## Gene expression profiling

Total RNA was extracted using TRI reagent from isolated BMDMs treated with LPS or pipecolic acid. Then RNA samples were sent to Gene Denovo Biotechnology Co. (Guangzhou, China) for further

sequencing. Briefly, eukaryotic mRNA was enriched by Oligo (dT) beads and fragmented into short fragments using fragmentation buffer and then reverse transcribed into cDNA. CDNA of about 200 bp was selected with AMPure XP beads, amplified through PCR, and purified with AMPure XP beads to construct cDNA library. RNA quality was assessed using an Agilent 2100 Bioanalyzer (Agilent Technologies, Palo Alto, CA, USA) and checked using RNase free agarose gel. The cDNA products were size selected by agarose gel electrophoresis, PCR amplified, and sequenced using Illumina Novaseq6000. Bioinformatic analysis was conducted on Omicsmart platform (https://www.omicsmart.com/).

### Primary hepatocytes isolation and culture

Primary hepatocytes were isolated by two-step in situ collagenase perfusion as described in our previous study[41]. In brief, 8-week-old mice were anesthetized and the liver was perfused in situ via the portal with Krebs Ringer Buffer (KRB, 118 mM NaCl, 4.7 mM KCl, 2.5 mM $CaCl_2$, 1.2 mM $MgCl_2$, 25 mM $NaHCO_3$, 10 mM glucose) with 1.7 mM EDTA for less than 10 min, while the inferior vena cava was used as outflow. Next, the perfusate was switched to digestion buffer (50 ml KRB buffer containing 45 mg type IV collagen and 3 ml of 125 mM $CaCl_2$) for 10 min. After digestion, the liver was acquired and the cell suspension was filtered. Two additional washing steps with KRB buffer were performed. Then a volume of 4 ml cold Percoll solution (9 parts Percoll to 1 part 1.5 M NaCl, pH 5–5.5) was used for every 6 ml of cell suspension (5 million cells/ml) and centrifuged (100 g, 4 °C, 10 min). The cells were then cultured with DMEM containing 10% FBS and 1% penicillin-streptomycin.

### Viral transfection and RNA interference

Sed and Exe mice received a single bolus tail vein injection of 100 ul AAV8 containing $1.3*10^{12}$ viral genomic (vg) (Orbitalgene Biotechnology) either carrying shRNA targeting Crym (Crym-shRNA) or scramble control (scramble-shRNA) at the age of 14 months. The targeting sequence for Crym knockdown is TCTTGTATCTCTCTGAAATAA. After 4 weeks of injection, mice were subjected to a single i.p. injection of LPS for sepsis induction. In another series of experiments, primary mouse hepatocytes were isolated and infected with SETD1A-overexpressing adenovirus (Ad-SETD1A) or a control Ad-eGFP constructed by Orbitalgene Biotechnology at a multiplicity of infection (MOI) of 50. For RNA interference, mouse primary hepatocytes were isolated and transfected with siRNA against SETD1A (si-SETD1A) or non-targeting control siRNA (si-Ctrl) by using the Lipofectamine RNAiMAX (Invitrogen). Sequences of siRNA against SETD1A are as following: #1, sense: 5'-GCAAGAAGGAGAAAGAAATT-3', anti-sense: 5'-UUAUCUUUCUCCUUCUUGCUT-3'; #2, sense: 5'-GGGCAAGACCCAAGGCAAATT-3', anti-sense: 5'-UUUGCCUUGGGUCUUGCCCTT-3'; #3, sense: 5'-GUUCUAUAUUGGACAGAUATT-3', anti-sense: 5'-UAUCUGUCCAAUAUAGAACTT-3'.

### DNA methylation assay and whole genome bisulfite sequencing (WGBS)

The global DNA methylation level of liver tissue was measured by Gene Denovo Biotechnology Co (Guangzhou, China). Briefly, genomic DNA from liver tissue was extracted by DNeasy Blood & Tissue Kit (Qiagen, 69506), randomly interrupted into 100–300 bp by Sonication (Covaris, MA, USA) and purified with MiniElute PCR Purification Kit (QIAGEN, MD, USA). Then, the fragmented DNAs were repaired by adding a single "A" nucleotide to the 3' end of blunt fragments, and the genomic fragments were ligated to methylated sequencing adapters. After that, bisulfite treatment was performed with Methylation-Gold kit (ZYMO, CA, USA), by which the unmethylated cytosine was converted to uracil. Finally, the converted DNA fragments were PCR-amplified and sequenced using Illumina HiSeqTM 2500. To assess different methylation patterns in different genomic regions, the methylation profile was plotted based on the average methylation levels.

### ChIP-seq

ChIP-seq were performed by SeqHealth Co (Wuhan, China). Briefly, cells were crosslinked with 1% formaldehyde for 10 min at room temperature and quenched with 125 mM glycine. The cells were treated with cell lysis buffer and nucleus was collected and sonicated to fragment chromatin DNA. The fragmented chromatin fragments were pre-cleared and then immunoprecipitated with anti-H3K4me3 (#9751, Cell Signaling Technology) antibodies. The DNA of input and IP was extracted by phenol-chloroform method. The high-throughput DNA sequencing libraries were prepared by using VAHTS Universal DNA Library Prep Kit for Illumina V3 (Catalog NO. ND607, Vazyme). The library products corresponding to 200–500 bps were enriched, quantified and finally sequenced on Novaseq 6000 sequencer (Illumina) with PE150 model.

Raw sequencing data was first filtered by Trimmomatic (version 0.36), low-quality reads were discarded and the reads contaminated with adapter sequences were trimmed. The clean reads were mapped to the mouse genome using STAR software (version 2.5.3a). The RSeQC (version 2.6) was used for reads distribution analysis. The MACS2 software (Version 2.1.1) was used for peak calling.

### ChIP

The ChIP assay was performed with the SimpleChIP Enzymatic Chromatin IP Kit #9002 (Cell Signaling Technology, Danvers, MA, USA) according to the manufacturer's instructions, and all reagents unless otherwise specified were provided in the kit. Briefly, primary hepatocytes were isolated from sedentary or exercised mice and cross-linked them with 1% formalin at room temperature. The chromatin was sheared by sonication. DNA fragments were incubated with anti-H3K4me3 (#9751, Cell Signaling Technology), positive control anti-H3 (#4620, Cell Signaling Technology) or negative control normal rabbit IgG (#2729, Cell Signaling Technology) antibodies. overnight at 4°C. ChIP-grade protein G magnetic beads were added to the immune complexes, the beads were washed, and the immunoprecipitates were eluted with an elution buffer contained in the kit. Following reversal of protein-DNA cross-links, DNA was purified using spin columns provided in the CHIP kit and dissolved in TE buffer. The enriched DNA fragments in ChIP were quantified with qPCR. Primer sequences used in qPCR for ChIP are as following: p (-1045/-882) F: 5'-GCCCCAGAGGAGAAGAGACT-3', R: 5'-CTGCCTACAGTGCTGCTGAG-3'; p (-883/-719) F: 5'-GAGAGCGCAATTTCAACTCC-3', R: 5'-TGAGCTGTGTGATGGCTTTC-3'; p (-577/-455) F: 5'-TTTCCAAGGACAGGGAGTTG-3', R: 5'-GAGGGCTTGTTGATCTCTGC-3'; p (-366/-190) F: 5'-GTCGGGACCTTTGGAGAAGT-3', R: 5'-CAGAGGAGGTGCAGGATCAC-3'; p (-123/ + 38) F: 5'-CAGGAGCTGAGCCCTAAATG-3', R: 5'-CCGCCTTGATTCCAATCTTA-3'.

### Statistics

The data in most figure panels reflect experiments performed using independent samples. Statistical analyses were conducted using SPSS software (version 26.0). All data are presented as the mean ± SEM values. The normality of distribution was assessed by using the Shapiro−Wilk test. Difference between two groups were assessed by unpaired two-tailed Student's t-test or Mann-Whitney test when appropriate. Difference between multiple groups were assessed by one-way ANOVA, two-way ANOVA, or two-way repeated measures ANOVA using the Bonferroni multiple comparison test. A $p$-value of less than 0.05 was considered statistically significant.

### Reporting summary

Further information on research design is available in the Nature Portfolio Reporting Summary linked to this article.

## Data availability

RNA sequence data that support the findings of this study have been deposited in GSA with the accession code "CRA011636". Chip-Seq data generated in this study have been deposited in the Gene Expression Omnibus database under accession code "GSE255161". The metabolome data generated in this study have been deposited in the OMIX, under accession code "OMIX006004". All other data are available within the Article and Supplementary Files, or available from the corresponding authors on request. Source data are provided with this paper.

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

## Acknowledgements

This work was supported by the grants from the Key Project of National Natural Science Foundation of China (31930055 for F.G.), National Key Basic Research Program of China (2020YFC2002900 for F.G.), National Natural Science Foundation of China (32271150 and 32071108 for J.L.; 32071169 for X.Z.; 82304165 for N.Z.), Shaanxi Province Outstanding Youth Fund of China (2022JC-15 for J.L.), Key R&D Plan of Shaanxi Province (2023-YBSF-535 for J.L.), and Foundation Grants of the Fourth Military Medical University (2022YBZC05 and 2021ZTXM-005 for N.Z.).

## Author contributions

F.G., J.L., and X.Z. conceived and supervised the study; J.L., X.Z., N.Z., and F.G. designed the experiments; N.Z., X.W., M.F., M.L., J.W., H.Y., Z.X. and S.H. performed the experiments; N.Z., X.W., M.F., M.L., J.W., Z.X., S.H., L.S., X.J., M.S., Y.W. and C.R. analyzed the data. N.Z., X.W., J.L., and F.G. wrote and revised the manuscript. N.Z. and X.W equally contributed to this study.

## Competing interests

The authors declare no competing interests.
