## [Peer Review File · Nature Communications]

Early-life exercise induces immunometabolic epigenetic modification enhancing anti-inflammatory immunity in middle-aged male miceReviewer #1 (Remarks to the Author):

This paper shows a thorough and dedicated analysis of the role of pipecolic acid in exercise-induced long-lasting anti-inflammatory effects. Pipecolic acid production is induced by exercise and remains upregulated even after a long period of detraining, resulting in decreased cytokine responses due to inhibition of the mTOR pathway.

The paper is clearly written and the experiments are well designed, but of course I have some comments, questions, and notes on inconsistency:

- The title should be adjusted: "anti-infection immunity" should be rephrased and the title should mention that this paper consists of murine (thus not human) data.
- The abstract should be more specific: words as "until later life", "enhanced immune defense", "late adulthood", "anti-infection immunity" should be specified, e.g. which age and what type of immune defense. Furthermore it should be added on which type of material the metabolic analysis was performed and the origin of the mentioned macrophages should be specified (BMDM, peritoneal, alveolar?). Furthermore for clarity it could be added that piper colic acid is a non-encoded amino acid.
- Line 63: "in mice" should be added.
- Figure 1A (and further figures): Add capitals at the beginning of each line (e.g. White)
- Figure S1C: this is non-LPS-stimulated, right? How do the authors explain such high IL1b levels? Usually in healthy mice no IL1b is detected in peripheral blood.
- Line 95: change into "To evaluate the immune response in acute infection"
- Fig 1C,D + fig 3I,J: also provide the absolute weight and temperature, so state that there is no baseline difference between the two groups.
- Fig 1K,L: do the authors have an explanation why there is baseline IL1b production (without LPS) and why the IL1RA levels are relatively low (usually about 5-10x higher than IL1b)?
- Line 107-109: Fig 1F-I show absolute white blood cell count, they don't provide any evidence of immunoparalysis (e.g. reduced cytokine production upon restimulation)
- Fig S3K,L: why is IL1b not shown here?
- Line 119: as stated above, immunoparalysis has not been tested here as not restimulation was performed
- Line 128-129: there are no signs presented of an enhanced immune response by exercise, only a decreased immune response. Based on which experimental finding is this claim made? Moreover it should be noted that in these experiments sepsis was induced by LPS and not by a live pathogen. With a live pathogen an enhanced initial immune response would lead to lower cytokine responses later, as more bacterial would be killed and they don't replicate. This is of course not the case for LPS.
- Line 130: same for "improved immune response".
- Line 136: add on which material (peripheral blood, bone marrow, liver?) the analysis was performed.
- Table S2,S3: are the mentioned P-values adjusted for multiple testing? (especially as only 1 metabolite was found). This should be added to the legend/table.
- Line 140: add that the analysis was performed in serum
- Figure 2I: are these ratios? What is the difference between concentration (J) en levels (I)? The panel of legend should mention that these are human data.
- Line 146: should the ref be Table S4, instead of S3?
- Line 154: specify the age of these youths?
- Figure 3B: please specify the time point
- Line 169: Fig3C-F don't show WBC responses, they only show absolute numbers. No claim on immune paralysis could be made on this.
- Fig 3K,L: why was IL1b not assessed here?
- Line 198-202 + Fig4E-G: the claim that rapamycin abolishes the effects of pipecolic acid cannot be made. They only effect that is shown here is that both rapamycin and pipecolic acid abolish the effect of LPS. Combination of rapamycin and pipecolic acid does not result in extra effect of abolishment.
- Fig S5: why did the authors choose to evaluate the mTOR pathway and not the effect on the TNF pathway? Assessment of H3K4me3 of the promotor of TNFA would be interesting to assess as well, as

a direct effect of pipercolic acid on TNFA is also possible.

-Fig 5B,C: specify the time point in the legend

-Fig 5O,P: would be interesting to see IL1b levels too, in order to determine the specificity for the TNF pathway (especially given the results of FigS5)

-Fig 6: in order to prove the causal role of Setd1a an experiment with an inhibitor of Setd1a is essential to show that this is not just a bystander effect.

-Line 253: there is no enhanced, but a decreased immune response to LPS (cytokine levels are lower, not higher)

-Line 264: ref 28 and 29 don't show a "strengthened immune response", they merely show fewer infections, of infections with fewer symptoms. Whether that is due to a modulation of the immune response was not assessed in these studies. Potential it could also be due to a weaker immune response, as less inflammation causes less complaints.

-Line 275: "a reduction of total leukocytes": both figure 1 and 3 do not show baseline reductions on leukocytes. Based on which data this claim is made?

-Line 279: which response is meant here? The cytokine response is decreased, not strengthened.

-Line 293-294: in human no data were shown that this lasted into adulthood after a long time of detraining. This should be corrected.

-In the discussion a section should be added on the following: In this paper sepsis was induced by LPS and not by a live pathogen. With a live pathogen an enhanced initial immune response would lead to lower cytokine responses later, as more bacteria would be killed and therefore they don't replicate. This is of course not the case for LPS. In sepsis there are two causes of death: 1. due to a too extreme initial cytokine response and therefore shock and multi-organ failure, or 2. due to a too weak initial cytokine response and therefore replication and dissemination of the bacterium. As in these experiment LPS was used to induced sepsis, only claims on point 1 can be made. Point 2 is left out of consideration in these experiments.

Reviewer #2 (Remarks to the Author):

The data from these studies are intriguing and novel, and support the value of exercise early in life in improving immune responses later in life via liver pipercolic acid production. The research designs of the animal and human studies in this paper are sound, and support the conclusions drawn from the data. The paper is well-written and follows a logical sequence. The data significantly advance scientific understanding in the area of exercise immunology. This reviewer has only a few minor considerations.

1. 15-month old mice are regarded as "middle-aged". Please defend the use of "late adulthood" in your title and other sections of the paper.

2. Lines 331-334, human studies: Reference #13 should be noted here. More information should be given (e.g., exercise duration of 20 minutes, workload of 100 W (male)/70 W (female)). This reviewer went to reference #13 and could not find data on pipercolic acid in this paper. Did you conduct an additional analysis (Figure 2I)? ALSO, why did you not compare groups of middle-aged adults to confirm your findings of elevated pipercolic acid in early-life trained 15-month old mice? Can your data in mice be extended to humans without that information? You may need to list this as a limitation in the discussion.

3. Line 163: Can you provide more information on your decision to use 200 micrograms / kg pipercolic acid (administered intraperitoneally one hour prior to and 24 hours following LPS injection in mice)?

Reviewer #3 (Remarks to the Author):

Zhang et al. demonstrate that a short period of exercise training early in life enhances immune responses to LPS after a prolonged detraining period in mice. They provide evidence that pipercolic acid

is a key mediator of this response, and support an exercise effect on increasing pipecolic acid in humans using an existing cross-sectional cohort. This is novel, although metabolomic screens (properly referenced by the authors) have previously identified this molecule as potentially related to beneficial aspects of exercise.

This study supports a long literature demonstrating that exercise training potentiates immune responses. Mechanisms underlying exercise effects are not well understood, so this paper represents an important advance in this respect. The most interesting results in my mind are the persistence of the effect after a significant detraining period.

This is a well-written paper. I had the following comments:

- 1) Line 92, a conclusion that chronic low-grade inflammation is affected by this training protocol is not supported by the design, since there is no evidence that inflammation here is abnormal (and this is unlikely in young healthy mice).
- 2) For murine sepsis scoring, please indicate rater blinding to treatment (if any).
- 3) Line 103, a multiple organ injury score is listed without context, and seems to be referring to MSS.
- 4) Interpretations of some of the data are problematic or incomplete, especially in the shRNA mouse experiments and similar. No statistical comparisons appear to be made between, e.g., scramble-shRNA and Crym-shRNA groups, and these comparisons are necessary to conclude (for example) that "inhibition in the liver abolished the benefits of early-life exercise against LPS" (line 224-225). Better statistical design/analysis is needed to draw some of the indicated conclusions in experiments in Figure 5 especially. See also statistical comments below.
- 5) The conclusion that pipecolic acid production is mediated by H3K4me3 in the Crym promoter (line 255) is not fully justified by the data. Although the analyses suggest this is the case and it is a reasonable likelihood, the experiments are not definitive absent manipulation of this pathway.
- 6) The human data on pipecolic acid and exercise is a strength of this study. However, it does not fully support the long-term effects of early life exercise. This is ignored to some extent in the discussion, and some further effort should be made to address this and how it could be studied in the future.
- 7) Statistical design is a problem. Many of the experiments are paired/repeated measures designs, and it does not appear that the statistical analysis is appropriate for this. This obviously includes things like body weight kinetics in Figure 1. However, this also includes BMDM experiments in Figure 4, assuming that macrophages from the same mouse were split and treated with all doses of Pip, Rapa, LPS, etc. There are also many factorial designs which are not fully analyzed, and the authors appear to rely on simple t-tests and ANOVAs for selected main effects, while ignoring other potential main effects and interactions.
- 8) LPS-induced endotoxemia is a suboptimal model of sepsis in mice (doi: 10.1089/sur.2016.021) and should probably not be referred to as such. Some discussion of the limitations of this model, and how these limitations could be overcome in future studies (e.g., the use of a live infection model) would be useful.
- 9) The sole use of male mice and male human subjects is a limitation.

--Brandt Pence

Dear Reviewers,

We appreciate very much the insightful comments and constructive suggestions provided in your review, which have greatly helped to improve the quality of our manuscript. After careful consideration, my co-authors and I have addressed each question and suggestion point-by-point. We have conducted additional experiments, performed more detailed analysis of the results, added relevant references and improved the wording and expressions throughout the revised manuscript. All the changes are highlighted in red color in the revised manuscript. Here is the point-by-point response to reviewers' comments.

Point-by-point response to reviewers' comments

Reviewer 1

We thank the reviewer very much for investing valuable time and effort in reviewing our manuscript. Your expertise and meticulous evaluation have greatly helped improve the revised version of our manuscript.

This paper shows a thorough and dedicated analysis of the role of pipercolic acid in exercise-induced long-lasting anti-inflammatory effects. Pipercolic acid production is induced by exercise and remains upregulated even after a long period of detraining, resulting in decreased cytokine responses due to inhibition of the mTOR pathway.

The paper is clearly written and the experiments are well designed.

Thank the reviewer for your positive comments.

Q1. The title should be adjusted: "anti-infection immunity" should be rephrased and the title should mention that this paper consists of murine (thus not human) data.

A1. As per your suggestion, we have revised the title to "Early-life exercise induces immunometabolic epigenetic modification enhancing anti-inflammatory immunity in middle-aged male mice".

Q2. The abstract should be more specific: words as "until later life", "enhanced immune defense", "late adulthood", "anti-infection immunity" should be specified, e.g. which age and what type of immune defense. Furthermore it should be added on which type of material the metabolic analysis was performed and the origin of the mentioned macrophages should be specified (BMDM, peritoneal, alveolar?). Furthermore for clarity it could be added that piper colic acid is a non-encoded amino acid.

A2. Thank you for these valuable suggestions. We have made revisions in the abstract per your suggestions, using more specific expressions, i.e. anti-inflammatory immunity, middle age (according to the second reviewer's suggestion), curbed cytokine response and mitigated sepsis when exposed to LPS challenge, metabolomics analysis of serum and liver, bone marrow-derived macrophages, and pipecolic acid is a non-encoded amino acid, to improve the clarity in the revised manuscript.

Q3. *Line 63: "in mice" should be added.*

A3. Thank you. It has been added in the revised manuscript (Line 67).

Q4. *Figure 1A (and further figures): Add capitals at the beginning of each line (e.g. White)*

A4. We have added capitals at the beginning of each line in Figure 1A, Figure 3A, Figure 5D, Figure S3A.

Q5. *Figure S1C: this is non-LPS-stimulated, right? How do the authors explain such high IL1b levels? Usually in healthy mice no IL1b is detected in peripheral blood.*

A5. Yes, Fig.S1C is non-LPS-stimulated in Sed and Exe mice at different ages (1, 4 and 15 months old). Thank you for your expert comment regarding the level of IL-1 β in peripheral blood in healthy mice. We agree with you that the level of IL-1 β is usually low in the peripheral blood of healthy mice in many previous studies. While in our study, serum levels of IL-1 β in mice at baseline (around 100 pg/ml at 4 months old and 200 pg/ml at 15 months old) appeared to be higher than what was typically reported in the literature, except some studies reporting similar baseline levels of IL-1 β to what we have observed (*J Periodontal Res.* 2022 doi: 10.1111/jre.13062; *Food Funct.* 2023 Jan 3;14(1):335-343. doi: 10.1039/d2fo02775e; *Int J Mol Sci.* 2023 Jan 13;24(2):1583. doi: 10.3390/ijms24021583). This variance may be attributed to several factors, including differences in animal housing, experimental conditions, sampling technique, as well as variations in the brands and batches of assay kits used. In particular, in our study, male C57BL/6J mice in the exercise group (Exe) underwent swimming sessions in water at a temperature of 33-35°C for 60 min, 5 days a week. Following each swimming session, the animals were gently dried with a cloth towel. For the non-exercised mice (Sed), a brief swim for only 30 s was conducted each day, followed by similar drying procedures. These experimental conditions might cause differences between the mice in the present study and those in normal housing environment. In addition, the Mouse IL-1 β ELISA Kit (CSB-E08054m, CUSABIO, China, Web: <https://www.cusabio.com/>, <https://joplink.net/mouse-interleukin-1-il-1-elisa-kit-csb-e08054m/>) used in our study has the detection range

between 31.25 pg/ml and 2000 pg/ml with sensitivity of 7.8 pg/ml, which may also contribute to the variance. We have acknowledged and discussed this issue in the revised manuscript (Line 286-290)

Q6. Line 95: change into "To evaluate the immune response in acute infection"

A6. Thank you and we have made changes in the revised manuscript (Line 96).

Q7. Fig 1C,D + fig 3I,J: also provide the absolute weight and temperature, so state that there is no baseline difference between the two groups.

A7. Per your suggestion, we have provided absolute weight and temperature in the supplemental material of the revised version (Excel 1 and Excel 2).

Q8. Fig 1K,L: do the authors have an explanation why there is baseline IL1b production (without LPS) and why the IL1RA levels are relatively low (usually about 5-10x higher than IL1b)?

A8. Please refer to the answer to Q5 about the baseline IL-1 β level in the present study. We agree with you that serum IL-1Ra level is usually higher than that of IL-1 β . In a recent study by Siri Tahtinen (*Nat Immunol.* 2022;23(4):532-542. doi: 10.1038/s41590-022-01160-y), IL-1 and IL-1ra are key regulators of the inflammatory response to RNA vaccines and the regulatory role of IL-1ra could be generalized to other types of innate immune stimulation including LPS. Their study demonstrated a markedly greater induction of IL-1ra compared to IL-1 α or IL-1 β , offering protection against uncontrolled systemic inflammation, while the "buffering capacity" of IL-1ra is likely to be overcome as the degree of innate stimulation is increased. In our study, we observed comparable levels of serum IL-1Ra and IL-1 β in mice, which might be attributed to high LPS dose used in our study (2 mg/kg body weight, i.e. 50 μ g per mouse, as opposed to 1 μ g per mouse in Siri Tahtinen's study), as well as potential variations in brands and batches of assay kits. Additionally, our data are also supported by some previous studies, in which the baseline serum IL-1Ra level in mice is comparable to baseline IL-1 β level (*Nat Metab.* 2021 Jun;3(6):843-858. doi: 10.1038/s42255-021-00402-x_Figure 1c; *J Affect Disord.* 2023 Aug 15;335:358-370. doi: 10.1016/j.jad.2023.05.049_Figure 1FI). We have acknowledged and discussed this issue in the revised manuscript (Line 286-296).

Q9. Line 107-109: Fig 1F-I show absolute white blood cell count, they don't provide any evidence of immunoparalysis (e.g. reduced cytokine production upon restimulation)

A9. Thank you for your expert comments. We agree with your suggestion that confirming immunoparalysis through reduced cytokine production upon restimulation would provide stronger evidence. In reference to the study by Napier BA, et al. (*Proc Natl Acad Sci U S A. 2019 Feb 26;116(9):3688-3694. doi: 10.1073/pnas.1814273116*), which defines “sepsis-associated immunoparalysis” as the persistence of a suppressed immune state during sepsis, we initially used the term “immunoparalysis” to describe the state of mice following LPS-induced sepsis, characterized by low body temperature, low blood sugar and low white blood cell counts.

In response to your expert suggestion, we have removed the term “immunoparalysis” from the revised manuscript.

Q10. Fig S3K,L: why is Il1b not shown here?

A10. Thank you for your question. The data of IL1b was not initially included in the original Fig S3K because of the constraints in the figure’s layout and limited space. Per your suggestion, we have now included the data of IL-1 β as in Figure S3L of the revised manuscript.

Q11. Line 119: as stated above, immunoparalysis has not been tested here as not restimulation was performed.

A11. We agree to the reviewer’s expert comment and have removed the term “immunoparalysis” from the revised manuscript.

Q12. Line 128-129: there are no signs presented of an enhanced immune response by exercise, only a decreased immune response. Based on which experimental finding is this claim made? Moreover it should be noted that in these experiments sepsis was induced by LPS and not by a live pathogen. With a live pathogen an enhanced initial immune response would lead to lower cytokine responses later, as more bacterial would be killed and they don't replicate. This is of course not the case for LPS.

A12. Thank you very much for your expertise comments. We have replaced “enhanced immune response by exercise” with “enhanced anti-inflammatory immunity, as evidenced by restrained cytokine response triggered by LPS, reduced tissue damage and improved recovery from sepsis” in the revised manuscript (Line 122-124).

In addition, we have taken note of your point regarding the limitations of the LPS-induced sepsis model compared to infections caused by live pathogens. The limitation of LPS-induced sepsis model has been added in the Discussion of revised manuscript (Line 328-332).

Q13. Line 130: same for "improved immune response".

A13. Thank you and we have replaced the expression of "improved immune response" with "enduring immunoregulating benefit" in the revised manuscript (Line 124).

Q14. Line 136: add on which material (peripheral blood, bone marrow, liver?) the analysis was performed.

A14. We have added "the serum and the liver" in the revised manuscript (Line 130).

Q15. Table S2,S3: are the mentioned P-values adjusted for multiple testing? (especially as only 1 metabolite was found). This should be added to the legend/table.

A15. Thank you for this valuable suggestion. In Table S2 and S3, the *P*-values are not adjusted for multiple testing, while we have provided false discovery rate (FDR) in the revised manuscript, which is adjusted for multiple testing.

Q16. Line 140: add that the analysis was performed in serum

A16. Thank you. We have added "in the serum of" in the revised manuscript (Line 134).

Q17. Figure 2I: are these ratios? What is the difference between concentration (J) en levels (I)? The panel of legend should mention that these are human data.

A17. We apologize for missing the unit in Y-axis of Figure 2I. The data are not ratios. They are original data obtained from targeted LC-MS with the unit of $\mu\text{mol/L}$. To ensure consistency, we have provided the calculated data converting the unit to ng/ml in Fig. 2I, aligning it with the units used in Fig. 2J in the revised manuscript. In addition, "Human data" has been indicated in the figure legend.

Q18. Line 146: should the ref be Table S4, instead of S3?

A18. We are sorry for this typo and we have made correction in the revised manuscript (Line 140).

Q19. Line 154: specify the age of these youths?

A19. The age of these youths (22.85 ± 3.40 years old) has been specified in the revised manuscript (Line 146).

Q20. Figure 3B: please specify the time point

A20. We have added the time point (immediately before LPS injection) in the revised legend of Figure 3B.

Q21. Line 169: Fig3C-F don't show WBC responses, they only show absolute numbers. No claim on immune paralysis could be made on this.

A21. As suggested by the reviewer, we have deleted “immunoparalysis” in the revised manuscript.

Q22. Fig 3K,L: why was IL1b not assessed here?

A22. Thank you for your comment. The level of IL-1 β was measured and we have now included the data of IL-1 β as in Figure 3L in the revised version.

Q23. Line 198-202 + Fig4E-G: the claim that rapamycin abolishes the effects of pipecolic acid cannot be made. They only effect that is shown here is that both rapamycin and pipecolic acid abolish the effect of LPS. Combination of rapamycin and pipecolic acid does not result in extra effect of abolishment.

A23. Thank you for your expert comment. We have changed into “the combination of mTORC1 inhibitor rapamycin and pipecolic acid did not exert further inhibitory effect on pro-inflammatory cytokines and iNOS in LPS-stimulated macrophages” in the revised manuscript (Line 190-192).

Q24. Fig S5: why did the authors choose to evaluate the mTOR pathway and not the effect on the TNF pathway? Assessment of H3K4me3 of the promotor of TNFA would be interesting to assess as well, as a direct effect of pipecolic acid on TNFA is also possible.

A24. Thank you for your insightful suggestion regarding the evaluation of the TNF pathway and the assessment of H3K4me3 of the promoter of TNFA. We fully agree on the potential direct effect of pipecolic acid on TNFA.

In our previous work on macrophages, we have found the pivotal role of mTORC1 activation in lysosomal biogenesis and cytokine production following LPS stimulation (Myeloid folliculin balances mTOR activation to maintain innate immunity homeostasis. *JCI Insight*. 2019;5(6):e126939. doi:10.1172/jci.insight.126939). Based on these findings, we conducted RNA-sequencing in LPS-stimulated BMDMs treated with or without pipecolic acid in the present study. Through KEGG analysis, we found that the mTOR signaling

pathway was enriched among the 324 differentially expressed genes (Figure S5C). Furthermore, GSEA analysis revealed that the mTORC1 pathway was activated by LPS stimulation but inhibited by pipecolic acid treatment (Figure S5D). Therefore, we focused on the mTOR pathway in the subsequent experiments.

We sincerely appreciate your suggestion about TNF- α and assessment of H3K4me3 of the promoter of TNF- α . This is an interesting issue warranting future studies. As suggested by the reviewer, we have included a discussion of these points in the revised manuscript (Line 318-320).

Q25. Fig 5B,C: specify the time point in the legend

A25. Thank you and the specific time point (mice at age of 15 months) has been indicated in the legend of Figure 5B, C in the revised manuscript.

Q26. Fig 5O,P: would be interesting to see IL1b levels too, in order to determine the specificity for the TNF pathway (especially given the results of FigS5)

A26. Thank you and we have provided the data of IL-1 β in Figure 5P of the revised manuscript.

Q27. Fig 6: in order to prove the causal role of Setd1a an experiment with an inhibitor of Setd1a is essential to show that this is not just a bystander effect.

A27. Thank you for your valuable suggestion. We agree with you that an experiment with modulation of Setd1a is essential to prove the causal role of Setd1a. Therefore, we performed additional experiments with SETD1A overexpression by adenovirus (Ad-SETD1A) transfection and SETD1A knockdown by siRNA in primary mouse hepatocytes. The results showed that overexpression of SETD1A led to increased H3K4me3 occupancy at Crym promoter (Figure 6J), elevated Crym mRNA and protein expressions (Figure 6G-I) in the hepatocytes. Conversely, knockdown of SETD1A decreased H3K4me3 occupancy at Crym promoter (Figure 6M), reduced Crym mRNA and protein expressions in the hepatocytes (Figure 6K-L). Taken together, these data have indicated that Setd1a-mediated H3K4me3 epigenomic modification contributes greatly to elevation of Crym expression in the liver. Since we mainly manipulated this pathway in *in vitro* experiments, we further added it as a limitation in the Discussion of revised manuscript (Line 333-336).

The relevant methods and results have been incorporated in the revised manuscript (Line 474-500; Line 232-250; Figure 6).

Q28. Line 253: there is no enhanced, but a decreased immune response to LPS (cytokine levels are lower, not higher)

A28. Thank you for your expertise comment. We have replaced the expression “enhanced immune response” with “curbed inflammatory response to LPS challenge and mitigated sepsis” in the revised manuscript (Line 254-255).

Q29. Line 264: ref 28 and 29 don't show a "strengthened immune response", they merely show fewer infections, of infections with fewer symptoms. Whether that is due to a modulation of the immune response was not assessed in these studies. Potential it could also be due to a weaker immune response, as less inflammation causes less complaints.

A29. Thank you for your valuable suggestion. We have deleted the expression "strengthened immune response" and changed into “moderate exercise programs help protect against acute respiratory infection” (Line 264-265).

Q30. Line 275: "a reduction of total leukocytes": both figure 1 and 3 do not show baseline reductions on leukocytes. Based on which data this claim is made?

A30. Figure S3F shows that the total leukocyte number was lower in mice after three-month swim training compared with sedentary control at baseline (without LPS stimulation). We have made it clear in the revised manuscript (Line 281-282, Fig S3F).

Q31. Line 279: which response is meant here? The cytokine response is decreased, not strengthened.

A31. We have deleted the inaccurate expressions and changed into “mice subjected to 3-month swimming training had reduced WBC number but a significant increase of WBC number when exposed to LPS” in the revised manuscript (Line 278-279).

Q32. Line 293-294: in human no data were shown that this lasted into adulthood after a long time of detraining. This should be corrected.

A32. Thank you for pointing this out. We have made it clear that the long-term effect of exercise was only investigated in a murine model in the present study and our data is limited to make a conclusion that early-life exercise induces long-lasting immune benefits in human. We have rephrased that statement and deleted “human” in the revised manuscript (Line 321-327).

Q33. In the discussion a section should be added on the following: I this paper sepsis was induced by LPS and not by a live pathogen. With a live pathogen an enhanced initial immune response would lead to lower cytokine responses later, as more bacteria would be killed and therefore they don't replicate. This is of course not the case for LPS. In sepsis there are two causes of death: 1. due to a too extreme initial cytokine response and therefore shock and multi-organ failure, or 2. due to a too weak initial cytokine response and therefore replication and dissemination of the bacterium. As in these experiment LPS was used to induced sepsis, only claims on point 1 can be made. Point 2 is left out of consideration in these experiments.

A33. Thank you very much for your expertise and valuable suggestions. As per your suggestion, we have added this in the discussion in the revised manuscript (Line 327-332):

“The murine sepsis model in this study was induced by LPS but not a live pathogen. LPS is a single component released by gram negative organisms and thus it neglects the host-pathogen interactions of gram-positive organisms and polymicrobial sepsis. Therefore, to better explore the immunological benefits of exercise, a more appropriate animal model, such as bacterial injection or cecal ligation and puncture is suggested to be used in the future study.”

Reviewer 2

We thank the reviewer very much for investing valuable time and effort in reviewing our manuscript. Your expertise and meticulous evaluation have greatly helped improve the revised version of our manuscript.

The data from these studies are intriguing and novel, and support the value of exercise early in life in improving immune responses later in life via liver pipecolic acid production. The research designs of the animal and human studies in this paper are sound, and support the conclusions drawn from the data. The paper is well-written and follows a logical sequence. The data significantly advance scientific understanding in the area of exercise immunology.

Thank the reviewer for your positive comments. We are delighted that you found our findings intriguing and novel.

Q1. 15-month old mice are regarded as "middle-aged". Please defend the use of "late adulthood" in your title and other sections of the paper.

A1. Thank the reviewer for the constructive suggestions. We have replaced the expression of "late adulthood" with "middle age, middle-aged phase" in the title and other sections of the revised manuscript.

Q2. Lines 331-334, human studies: Reference #13 should be noted here. More information should be given (e.g., exercise duration of 20 minutes, workload of 100 W (male)/70 W (female)). This reviewer went to reference #13 and could not find data on pipecolic acid in this paper. Did you conduct an additional analysis (Figure 2I)? ALSO, why did you not compare groups of middle-aged adults to confirm your findings of elevated pipecolic acid in early-life trained 15-month old mice? Can your data in mice be extended to humans without that information? You may need to list this as a limitation in the discussion.

A2. Thank you for your expert comment. Per your suggestion, Reference#13 has been noted in Line 351, and the detailed description of exercise protocol has been provided in the Methods of the revised manuscript (Line 353-355). In our previous study cited in Reference#13, elevated pipecolic acid was found in young healthy volunteers following a single bout of exercise (the fifth metabolite in Figure 3D, *Life Sci.* 2023;313:121284. doi: 10.1016/j.lfs.2022.121284). The data shown in Fig. 2I are original data from targeted LC-MS with the unit of $\mu\text{mol/L}$. We apologize for missing the unit in Y-axis and we have provided the calculated data converting the unit $\mu\text{mol/L}$ to ng/ml in Fig.2I, making it consistent with Fig. 2J in the revised manuscript.

Thank you for your valuable suggestions on the limitation of our study. The long-term effect of exercise was only investigated in a murine model in the present study. While we observed an increase in the circulating level of pipecolic acid in young human volunteers following a single bout exercise, and found higher levels in young highly-trained athletes, the available data is limited to draw a conclusion regarding the long-lasting immune benefits of early-life exercise in humans. Further studies should be planned to investigate the potential correlation between pipecolic acid and immune benefits in athletes during middle-age and detrained periods. We have added this as a limitation in the Discussion of the revised manuscript (Line 321-327).

Q3. Line 163: Can you provide more information on your decision to use 200 micrograms / kg pipecolic acid (administered intraperitoneally one hour prior to and 24 hours following LPS injection in mice)?

A3. We observed an anti-inflammatory effect of pipecolic acid at concentrations of 10 $\mu\text{mol/L}$ and 20 $\mu\text{mol/L}$ *in vitro*. Considering a mouse with a body weight of 25 g and an estimated blood volume of 2 ml, the appropriate dose of pipecolic acid for a mouse is approximately 5 μg , equivalent to 200 $\mu\text{g/kg}$ of body weight (the molecular weight of pipecolic acid is 129.16). Since, the half-life of pipecolic acid is about 12 hours, pipecolic acid was thus administered 25 hours later (which is 24 hours after LPS injection).

Reviewer 3

We thank the reviewer very much for investing valuable time and effort in reviewing our manuscript. Your expertise and meticulous evaluation have greatly helped improve the revised version of our manuscript.

Zhang et al. demonstrate that a short period of exercise training early in life enhances immune responses to LPS after a prolonged detraining period in mice. They provide evidence that pipecolic acid is a key mediator of this response, and support an exercise effect on increasing pipecolic acid in humans using an existing cross-sectional cohort. This is novel, although metabolomic screens (properly referenced by the authors) have previously identified this molecule as potentially related to beneficial aspects of exercise.

This study supports a long literature demonstrating that exercise training potentiates immune responses. Mechanisms underlying exercise effects are not well understood, so this paper represents an important advance in this respect. The most interesting results in my mind are the persistence of the effect after a significant detraining period.

This is a well-written paper.

Thank you for your positive comments and for recognizing the strengths of our manuscript. We are delighted to hear that you found our findings important and interesting.

Q1. Line 92, a conclusion that chronic low-grade inflammation is affected by this training protocol is not supported by the design, since there is no evidence that inflammation here is abnormal (and this is unlikely in young healthy mice).

A1. Thank you for your valuable suggestion. “chronic low-grade inflammation” was deleted in the revised manuscript.

Q2. For murine sepsis scoring, please indicate rater blinding to treatment (if any).

A2. Thank you and we have indicated that murine sepsis scoring was evaluated blindingly (the rater is blinded to treatment) in the Methods in the revised manuscript (Line 376).

Q3. Line 103, a multiple organ injury score is listed without context, and seems to be referring to MSS.

A3. Thank you for pointing this out. We apologize for the oversight and have made correction by replacing it with “a lower murine sepsis score (MSS)” in the revised manuscript (Line 104).

Q4. Interpretations of some of the data are problematic or incomplete, especially in the *shRNA* mouse experiments and similar. No statistical comparisons appear to be made between, e.g., scramble-*shRNA* and *Crym-shRNA* groups, and these comparisons are necessary to conclude (for example) that “inhibition in the liver abolished the benefits of early-life exercise against LPS” (line 224-225). Better statistical design/analysis is needed to draw some of the indicated conclusions in experiments in Figure 5 especially. See also statistical comments below.

A4. Thank you very much for your expertise comment and suggestion. Statistical comparisons have been made between scramble-*shRNA* and *Crym-shRNA* groups (Figure 5F-T) by using two-way ANOVA followed by Bonferroni’s test. We have labeled statistical symbol in Figure 5 ($^{\#}p < 0.05$, $^{\#\#}p < 0.01$ for interaction using two-way ANOVA) and indicated the specific statistical method in the legend of Figure 5 of the revised manuscript.

Q5. The conclusion that pipecolic acid production is mediated by H3K4me3 in the *Crym* promoter (line 255) is not fully justified by the data. Although the analyses suggest this is the case and it is a reasonable likelihood, the experiments are not definitive absent manipulation of this pathway.

A5. Thank you for your valuable comments. As suggested, we performed further experiments. ChIP-seq analysis revealed that H3K4me3 occupancy at the *Crym* promoter had a significant increase in hepatocytes of Exe mice compared with Sed mice aged at 15 months (Figure 6B). We further demonstrated that H3K4me3 occupancy at -366 ~ -190 bp of *Crym* promoter was increased in hepatocytes of both 4-month-old and 15-month-old Exe mice (Figure 6C-D). Then *Setd1a* was identified to be the only H3K4 methyltransferase that significantly increased in the hepatocytes of Exe mice compared to Sed mice both at the age of 4 months and 15 months (Figure 6E-F). To further address the reviewer’s concern, we performed additional experiments with SETD1A overexpression by adenovirus (Ad-SETD1A) transfection and SETD1A knockdown by siRNA in primary mouse hepatocytes. The results showed that overexpression of SETD1A led to increased H3K4me3 occupancy at *Crym* promoter (Figure 6J), elevated *Crym* mRNA and protein expressions (Figure 6G-I) in the hepatocytes. Conversely, knockdown of SETD1A decreased H3K4me3 occupancy at *Crym* promoter (Figure 6M), reduced *Crym* mRNA and protein expressions in the hepatocytes (Figure 6K-L). Taken together, these data have indicated that *Setd1a*-mediated H3K4me3 epigenomic modification contributes greatly to elevation of *Crym* expression in the liver. Since we mainly manipulated this pathway in *in vitro* experiments, we further added it as a limitation in the Discussion of revised manuscript (Line 332-335).

The relevant methods and results have been incorporated in the revised manuscript (Line 474-500; Line 512-525; Line 232-250; Figure 6).

Q6. The human data on pipecolic acid and exercise is a strength of this study. However, it does not fully support the long-term effects of early life exercise. This is ignored to some extent in the discussion, and some further effort should be made to address this and how it could be studied in the future.

A6. Thank you for your valuable comments on the strength and the limitation of our study. While we observed an increase in the circulating level of pipecolic acid in young human volunteers following a single bout exercise, and found higher levels in young highly-trained athletes, the available data is limited to draw a conclusion regarding the long-lasting immune benefits of early-life exercise in humans. Further studies should be planned to investigate the potential correlation between pipecolic acid and immune benefits in athletes during middle-age and detrained periods. We have added this as a limitation in the Discussion of the revised manuscript (Line 321-327).

Q7. Statistical design is a problem. Many of the experiments are paired/repeated measures designs, and it does not appear that the statistical analysis is appropriate for this. This obviously includes things like body weight kinetics in Figure 1. However, this also includes BMDM experiments in Figure 4, assuming that macrophages from the same mouse were split and treated with all doses of Pip, Rapa, LPS, etc. There are also many factorial designs which are not fully analyzed, and the authors appear to rely on simple t-tests and ANOVAs for selected main effects, while ignoring other potential main effects and interactions.

A7. Per your suggestion, we have consulted an expert in biomedical statistics from our university who assisted us in reviewing the statistical methods employed in the present study. Based on the expert's suggestion, we have clarified and specified all the statistical methods in the figure legends of the revised manuscript. Thank you for your valuable comments, which have contributed to enhancing the quality and integrity of the statistical analysis in our study.

Q8. LPS-induced endotoxemia is a suboptimal model of sepsis in mice (doi: 10.1089/sur.2016.021) and should probably not be referred to as such. Some discussion of the limitations of this model, and how these limitations could be overcome in future studies (e.g., the use of a live infection model) would be useful.

A8. Thank you for your suggestion. The sepsis mouse model in this study was induced by LPS but not a live pathogen. LPS is a single component released by gram-negative organisms and thus it neglects the host-pathogen interactions of gram-positive organisms and polymicrobial sepsis. Therefore, bacterial injection or cecal ligation and puncture model is suggested to be used in the future study. We have discussed this and added it as a limitation in the revised manuscript (Line 328-332).

Q9. The sole use of male mice and male human subjects is a limitation.

A9. Thank you very much for your valuable suggestions. The sole use of male mice has been indicated in the title, abstract and methods of the revised manuscript. In the single bout exercise experiment, a total of 27 young adults were recruited (aged 22.85 ± 3.40 years), of whom 18 adults are male and 9 adults are female. Per your suggestion, this has been clearly indicated in the methods of manuscript.

Yours Sincerely,

Feng Gao, MD, PhD
Professor of Physiology
Fourth Military Medical University
Xi'an 710032, China
Email: fgao@fmmu.edu.cn

Reviewer #1 (Remarks to the Author):

The authors have answered my questions, have performed additional experiments to enforce their data and improved the discussion. They have addressed all my raised points.

Reviewer #2 (Remarks to the Author):

The authors have responded appropriately to my review comments and made changes to the revised paper. I have no additional comments.

Reviewer #3 (Remarks to the Author):

The authors have addressed my previous concerns and comments. The new experiments and revised interpretations have substantially strengthened the manuscript.

Point-by-point response to reviewers' comments

Reviewer 1

The authors have answered my questions, have performed additional experiments to enforce their data and improved the discussion. They have addressed all my raised points.

Thank you for recognizing the improvements we have made to the manuscript based on your expert comments. We appreciate your time and consideration.

Reviewer 2

The authors have responded appropriately to my review comments and made changes to the revised paper. I have no additional comments.

Thank you for acknowledging our efforts in addressing your expert review comment. We appreciate your time and consideration.

Reviewer 3

The authors have addressed my previous concerns and comments. The new experiments and revised interpretations have substantially strengthened the manuscript.

Thank you for recognizing the improvements we have made to the manuscript based on your expert comments. We appreciate your time and consideration.

Yours Sincerely,

Feng Gao, MD, PhD
School of Aerospace Medicine, Fourth Military Medical University,
169 Changlexi Road, Xi'an 710032, China
Email: fgao@fmmu.edu.cn